# MosaicBERT: A Bidirectional Encoder Optimized for Fast Pretraining

**Jacob Portes**[1*]      **Alex Trott**[1*]      **Sam Havens**[1]      **Daniel King**[1]      **Abhinav Venigalla**[1]
jacob.portes@      alex.trott@      sam.havens@      daniel.king@      abhi@

**Moin Nadeem**[2†]      **Nikhil Sardana**[1]      **Daya Khudia**[1]      **Jonathan Frankle**[1]
moinnadeem@      nikhil.sardana@      daya.khudia@      jfrankle@

[1]MosaicML × Databricks
[1]@databricks.com  [2]@moinnadeem.com

## Abstract

Although BERT-style encoder models are heavily used in NLP research, many researchers do not pretrain their own BERTs from scratch due to the high cost of training. In the past half-decade since BERT first rose to prominence, many advances have been made with other transformer architectures and training configurations that have yet to be systematically incorporated into BERT. Here, we introduce MosaicBERT, a BERT-style encoder architecture and training recipe that is empirically optimized for fast pretraining. This efficient architecture incorporates FlashAttention, Attention with Linear Biases (ALiBi), Gated Linear Units (GLU), a module to dynamically remove padded tokens, and low precision LayerNorm into the classic transformer encoder block. The training recipe includes a 30% masking ratio for the Masked Language Modeling (MLM) objective, bfloat16 precision, and vocabulary size optimized for GPU throughput, in addition to best-practices from RoBERTa and other encoder models. When pretrained from scratch on the C4 dataset, this base model achieves a downstream average GLUE (dev) score of 79.6 in 1.13 hours on 8 A100 80 GB GPUs at a cost of roughly $20. We plot extensive accuracy vs. pretraining speed Pareto curves and show that MosaicBERT base and large are consistently Pareto optimal when compared to a competitive BERT base and large. This empirical speed up in pretraining enables researchers and engineers to pretrain custom BERT-style models at low cost instead of finetune on existing generic models. We open source our model weights and code.

## 1  Introduction

BERT has been the workhorse of modern natural language processing (NLP) since its introduction in 2018 [14]. Even in the era of large language models (LLMs), BERT-style encoder models are still quite relevant; for example, encoder models are used for vector database embeddings and retrieval augmented generation in tandem with LLMs [30, 35, 27, 61, 52]. In the past half-decade since BERT first rose to prominence, however, many advances have been made with other transformer architectures and training configurations that have yet to be systematically incorporated into BERT [12, 44, 11]. In this study we empirically show that these speed optimizations can successfully be incorporated into the classic BERT architecture and training recipe.

---

[*]equal contribution. Corresponding author jacob.portes@databricks.com. Code can be found at mosaicbert.github.io

[†]work done while at MosaicML.

37th Conference on Neural Information Processing Systems (NeurIPS 2023).

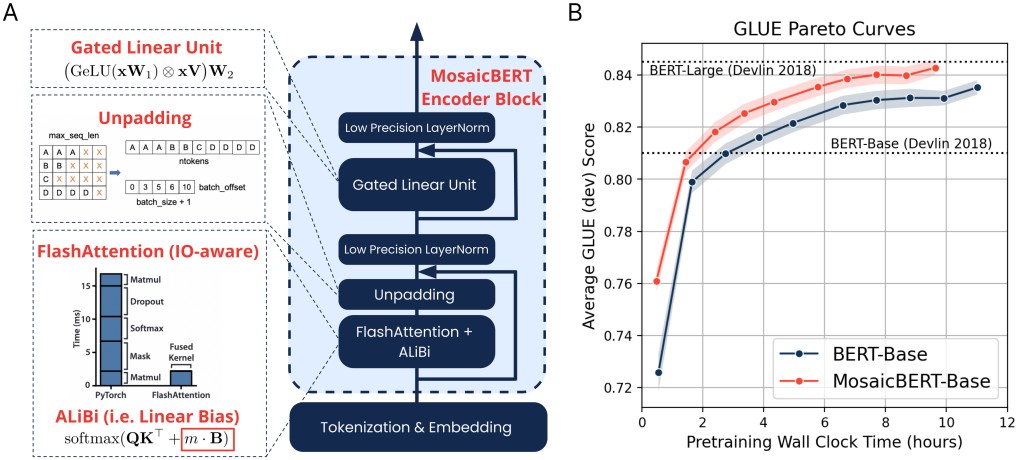

Figure 1: (A) Schematic of MosaicBERT architecture (B) Pareto curves of average GLUE (dev) scores for MosaicBERT-Base and the standard BERT-Base. Error bars indicate 95% confidence interval over n=5 pretraining seeds. All training was on $8\times$A100-80GB GPUs. FlashAttention schematic adapted from [11], and unpadding schematic adapted from [66]).

BERT-style models are typically trained in two stages: an initial self-supervised pretraining phase that builds general representations of language, and a subsequent supervised finetuning phase that uses those representations to address a specific task. The pretraining stage for BERT models has historically been computationally expensive; in the original BERT study, for example, the authors trained their models for 4 full days on 16 Google TPUs. In recent years, however, the time and cost to train BERT models has dropped significantly. One widely cited paper from 2021 successfully reduced the training time of BERT-Large to 24 hours on 8 Titan V-12 GPUs [28], and another very recent paper trained a competitive BERT-Base model on a single consumer GPU in only 24 hours [18]. Our work builds on these trends.

In this study, we introduce our optimized MosaicBERT architecture and show that certain architectural choices for BERT-style encoders lead to accuracy vs. time Pareto improvements in pretraining. We do this by empirically comparing MosaicBERT with an optimal BERT baseline that does *not* incorporate our architectural changes but *does* have non-architectural optimizations such as fast data streaming and optimal floating point precision. We then evaluate on the classic GLUE benchmark [60].

It can often be difficult to understand the effects of model architecture modifications and training hyperparameter choices by simply reporting a few numbers in a table, as is traditionally done in ML papers. A few numbers in a table can obscure differences in the training data, objective function, batch size, training duration, learning rate schedule and many other details. Since our main goal in this study is to show how combinations of architecture choices lead to improvements in both *accuracy* and *training time*, we make the choice to plot accuracy vs. training time Pareto curves for all models (for example, in Figure 1B). Certain architecture changes might lead to an increase in throughput, but a decrease in accuracy (e.g. changes to the floating point precision); other changes might lead to an increase in accuracy but take a much longer time to converge (e.g. increasing the model size). Accuracy vs. training time Pareto curves enable us to adequately asses all these changes (see Appendix for a more detailed discussion on Pareto curves).

The contributions of this work are as follows:

- We implement a new BERT-style encoder architecture that optimizes pretraining speed and accuracy. This architecture combines FlashAttention [11], ALiBi [44], Gated Linear Units [12, 50], a dynamic unpadding module [66], and low precision LayerNorm.

- We show that MosaicBERT base achieves the downstream average GLUE (dev) score of 79.6 in 1.13 hours on $8\times$A100 80 GB GPUs at a cost of roughly \$20 on a standard cloud provider.

- We characterize the accuracy vs. pretraining time Pareto frontier for MosaicBERT-Base and Large, and empirically show that the performance of MosaicBERT-Base and Large is Pareto optimal relative to BERT-Base and Large.
- We characterize the relative throughput properties of each of MosaicBERT's architecture and design choices via ablations.
- We characterize the tradeoff between model size and training duration, and show that BERT-Large performance only surpasses BERT-Base performance after extensive training. We open-source our model weights and code at `mosaicbert.github.io`.

Finally, we note that the ideas explored in this study are directly applicable to LLM pretraining, and directly motivated many of the architecture choices around MosaicML's MPT-7B and MPT-30B large language models [39, 38].

| Model Architecture | GLUE (dev) | Training (hours) | Hardware | Pretraining Corpus | Params |
|---|---|---|---|---|---|
| MosaicBERT-Base (ours) | 79.6 | 1.13 | 8×A100-80 | C4 | 137M |
| MosaicBERT-Base (ours) | 82.2 | 2.81 | 8×A100-80 | C4 | 137M |
| MosaicBERT-Base (ours) | 83.2 | 4.6 | 8×A100-80 | C4 | 137M |
| MosaicBERT-Base (ours) | 83.4 | 5.27 | 8×A100-80 | C4 | 137M |
| BERT-Base (our benchmark) | 83.2 | 11.5 | 8×A100-80 | C4 | 110M |
| BERT-Base [14, 18] | 81.0 | 96 | 16×TPU | Wiki+BookC | 110M |
| BERT-Base [18] | 73.6 | 24 | 1 RTX A6000 | Wiki+BookC | 110M |
| CrammingBERT-Base [18] | 80.4 | 24 | 1 RTX A6000 | Wiki+BookC | 110M |
| BERT-Large [14, 37] | 84.1 | 96 | 64×TPU | Wiki+BookC | 340M |

Table 1: Average GLUE (dev) score across various efficient BERT implementations. Average includes all 8 GLUE tasks. BERT-Base average GLUE scores on the dev set as reported by [18], Table 3. BERT-Large average GLUE scores on the dev set as reported by [37], Table 5. Note that the original training set for BERT was pretrained on 40 epochs of English Wikipedia and BookCorpus [68], while MosaicBERT was pretrained on the Colossal Cleaned Common Crawl (C4) corpus [46].

## 2 Methods

In order to build MosaicBERT, we incorporated architectural choices from the recent transformer literature. These include FlashAttention [11], ALiBi [44], training with dynamic unpadding [66], low precision LayerNorm, and Gated Linear Units [12, 50] (Figure 1). Before describing these modifications in detail, we first review the classic BERT architecture and how we chose a strong baseline.

Since our goal here is to show relative improvements in training time and final accuracy, we do not attempt to beat state of the art models for finetuning benchmarks such as GLUE [60]. These SOTA models are often trained for much longer (e.g. [37]) and are larger than the models we explore in this study [9, 23].

### 2.1 Choosing a Strong BERT Baseline

The basic transformer block used in BERT models consists of (1) the attention mechanism and (2) the feed forward layers. This block is then repeated depending on the model size; BERT-Base has 12 repeated transformer blocks, while BERT-Large has 24.

For our baseline BERT-Base, we used the exact architecture of BERT from [14];[3] this includes a hidden size of 768, an intermediate size of 3072, 12 attention heads and 12 hidden layers, as well as the GeLU activation function, and learned positional embeddings. For our baseline BERT-Large, we used the exact architecture of BERT-Large from [14], which has a hidden size of 1024, an intermediate size of 4096, 16 attention heads, and 24 hidden layers.

---

[3]The Hugging Face `bert-base-uncased` model has been downloaded 62 million times, more than any other model on the Hugging Face hub.

While MosaicBERT-Base (i.e. 12 hidden layers) and Large (i.e. 24 hidden layers) stay true to this general structure, we introduce modifications that affect both the attention mechanism and the feedforward layers.

## 2.2 MosaicBERT Architecture Modifications for Fast Pretraining

### 2.2.1 MosaicBERT Modifications to the Attention Mechanism

**FlashAttention**: The recently proposed FlashAttention layer reduces the number of read/write operations between the GPU HBM (high bandwidth memory, i.e. long-term memory) and the GPU SRAM (i.e. short-term memory) [11] (see also [45]). We modified the FlashAttention module built by Hazy Research with OpenAI's triton library in order to flexibly incorporate ALiBi [57].[4]

**Attention with Linear Biases (ALiBi)**: In most BERT models, the positions of tokens in a sequence are encoded using learned position embeddings, which are added to the learned token embeddings at the start of the forward pass. ALiBi eliminates position embeddings and instead encodes position information directly through the attention operation [44]. It adds a negative bias to the attention score between each token pair, which grows linearly with the relative distance between the tokens. Intuitively, this biases attention to nearby tokens and allows for extrapolation to context lengths longer than those used for training [44].

Following the notation in [44], the attention block computes the attention scores between the *i*th query $q_i \in \mathbb{R}^d$ and keys $\mathbf{K} \in \mathbb{R}^{L \times d}$ where $d$ is the head dimension and $L$ is the sequence length. ALiBi adds a fixed bias with $m$ as a head-specific slope controlling how the bias grows with absolute distance between tokens, yielding attention weights as

$$\mathtt{softmax}\big(q_i\mathbf{K}^\top - m \cdot \mathtt{abs}([i-1, i-2, ..., i-L])\big). \tag{1}$$

The slopes $m$ follow a geometric sequence such that for $n$ heads, each head has a ratio of $2^{-8/n}$ [44].

During finetuning or inference, the static bias can be increased to accommodated longer sequence lengths. For example, a model pretrained using ALiBi with a maximum sequence length of 128 tokens can then extrapolate to a task with 256 tokens with little to no decrease in zero-shot performance. This was the original motivation for AliBi (i.e. "train short, test long") [44]. Since pretraining a model with a maximum sequence length of 128 has much higher throughput than pretraining a model with a sequence length of 256, ALiBi can be considered an indirect speedup method.

### 2.2.2 Modifications to the Feedforward Layers

**Gated Linear Units (GLU)**: We used Gated Linear Units for the feedforward sublayer of a transformer. GLUs were first proposed in 2016 [12], and incorporate an extra learnable matrix that "gates" the outputs of the feedforward layer (Figure 1A). More recent work has shown that GLUs can improve performance quality in transformers [50, 42]. We used the GeLU (Gaussian-error Linear Unit)[5] activation function with GLU, which is sometimes referred to as GeGLU. The module can be described by the following equation

$$\big(\mathrm{GeLU}(\mathbf{x}\mathbf{W}_1) \otimes \mathbf{x}\mathbf{V}\big)\mathbf{W}_2, \tag{2}$$

where $\mathbf{x}$ is the input to the feedfoward layer and the matrix $\mathbf{V}$ gates the output of the GeLU activation function. The extra gating matrix in a GLU model potentially adds additional parameters to a model; we chose to augment our MosaicBERT-Base model with additional parameters due to GLU modules, as it leads to a Pareto improvement across all timescales. While BERT-Base has 110 million parameters, MosaicBERT-Base has 137 million parameters. Note that MosaicBERT-Base reaches higher accuracy faster than BERT-Base *despite having more parameters* (Figure 1B). Similarly, BERT-Large has 340 million parameters, and MosaicBERT-Large has 430 million parameters.

---

[4]`github.com/HazyResearch/flash-attention`. Note that while this research was being completed, PyTorch 2.0 was released with support for FlashAttention. However, FlashAttention in PyTorch 2.0 does not currently support ALiBi integration.

[5]GeLU is a fully differentiable approximation to ReLU, and was used in the original BERT study [14].

### 2.2.3 Additional Modifications

**Low Precision LayerNorm**: LayerNorm is a bandwidth-bound operation, which means that its speed depends on how quickly data can be loaded from memory to the compute units for element-wise operations. Typically the LayerNorm operation is set to `float32` precision, which requires 4-bytes per-element. In MosaicBERT, we modify LayerNorm modules to run in `bfloat16` precision instead of `float32`. This reduces the amount of data that needs to be loaded from memory, as only 2-bytes are required per-element. PyTorch's automatic-mixed-precision package does not, by default, run LayerNorm in lower precision because this can lead to numerical instabilities for certain models. However, our experimental results show that MosaicBERT does not experience any numerical instabilities with `bfloat16` precision LayerNorm.

**Unpadding**: Standard NLP practice is to combine text samples of different lengths into a batch, and pad the sequences with special padding tokens so that all sample sequence lengths are the same (Figure 1A). During training, however, this leads to many wasted operations on the padding tokens. In MosaicBERT, we take a different approach and instead concatenate all the examples from a minibatch into a single sequence of batch size 1. Results from NVIDIA[6] and others have shown that this approach leads to speed improvements during training, since operations are not performed on padding tokens [66].

**MLM Masking Ratio and Dropout**: We used the standard Masked Language Modeling (MLM) pretraining objective. While the original BERT paper also included a Next Sentence Prediction (NSP) task in the pretraining objective, subsequent papers have shown this to be unnecessary [37, 28]. For our BERT baselines, we used the standard 15% masking ratio. However, we found that a 30% masking ratio led to slight accuracy improvements in both pretraining MLM and downstream GLUE performance. We therefore included this simple change as part of our MosaicBERT training recipe. Recent studies have also found that this straightforward change can lead to downstream improvements [63, 1].[7]

For the baseline BERT, we applied the standard 0.1 dropout to both the attention and feedforward layers of the transformer block. For MosaicBERT, however, we applied 0.1 dropout to the feedforward layers but did not apply dropout to the FlashAttention module, as this was not possible with the OpenAI triton implementation [57].[8] The removal of dropout in the attention layers also leads to a small speed up.

## 2.3 Pretraining Optimizations for Both MosaicBERT and the BERT baseline

**Data**: Pretraining data is an important factor when comparing BERT models; while the original BERT study trained on English Wikipedia and BookCorpus [68], subsequent models such as RoBERTa trained on much larger datasets (e.g. [37] trained on 160 GBs of text while [14] only trained on 16GB of text). Here we chose to train all models on the more modern Colossal Cleaned Common Crawl (C4) corpus [46]. For all experiments, we used a maximum sequence length of 128 tokens. Note that in our implementation, samples longer than 128 tokens were simply truncated.

**Streaming Dataset**: As part of our efficiency pipeline, we converted the C4 dataset to the `StreamingDataset` format[9] and used this for both MosaicBERT-Base and the baseline BERT-Base. This ensured that our wall clock time measurements were not hindered by data streaming.

**Bfloat16 Precision**: We use `bfloat16` mixed precision training for all the models. `bfloat16` is a custom 16-bit floating point format for machine learning that has one sign bit, eight exponent bits, and seven mantissa bits, and has the dynamic range of `float32`. For mixed precision training, a matrix

---

[6]NVIDIA removes padding in their MLPerfv1.1 benchmarking results `https://github.com/mlcommons/training_results_v1.1` in the `padding.py` file

[7]In "Should You Mask 15% in Masked Language Modeling?" Wettig et al. find that constant MLM masking ratios above 15% lead to improved average GLUE and SQuAD scores for bert-base and bert-large. Similarly, in the recently published paper "Dynamic Masking Rate Schedules for MLM Pretraining," Ankner et al. find that a constant MLM masking ratio of 30% consistently outperforms a MLM masking ratio of 15% for bert-base.

[8]`github.com/openai/triton`

[9]`github.com/mosaicml/streaming`. It is a drop in replacement for PyTorch's `IterableDataset` that allows for fast streaming from cloud storage.

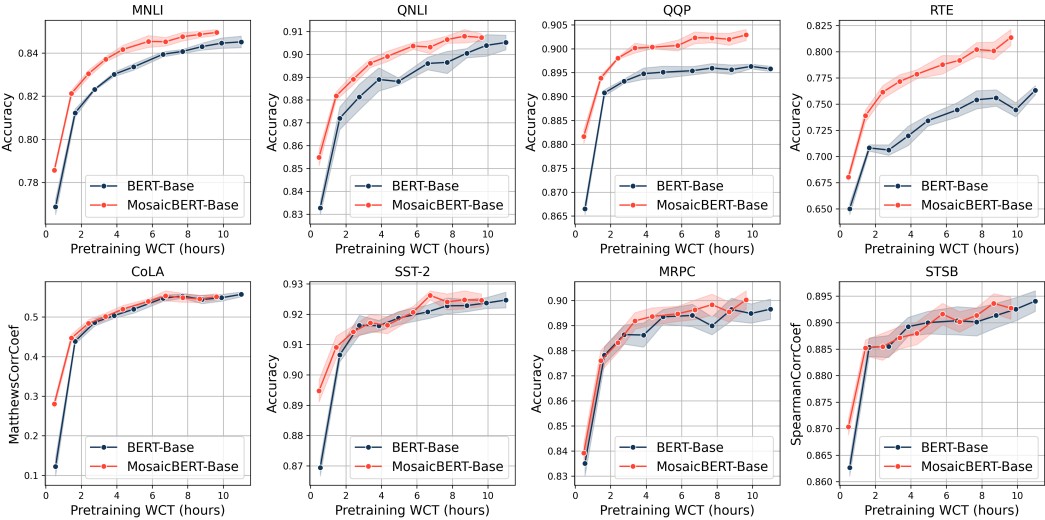

Figure 2: Performance on individual GLUE (dev) finetuning tasks. Our MosaicBERT-Base consistently outperforms BERT-Base on MNLI-m, QNLI, QQP and RTE, and has comparable performance on CoLA, SST-2, MRPC and STSB. Wall clock time is for $8\times$A100-80GB GPUs, and does not include finetuning time. Error bars are plotted with 95% confidence interval across $n = 5$ pretraining seeds, and all models are trained for 70,000 steps with batch size 4096.

multiplication layer uses `bfloat16` for the multiplication and 32-bit IEEE floating point (`float32`) for gradient accumulation. We found this to be more stable than using `float16` mixed precision.[10]

**Vocab Size as a Multiple of 64**: We increased the vocab size to be a multiple of 64 (i.e. from 30,522 to 30,528).[11] This small constraint is something established in the MEGATRON work by Shoeybi et al. [53], and leads to a non-trivial throughput speedup, as CUDA is more efficient at computing matrix multiplications with these dimensions.[12] Across all experiments, we use the standard BERT-Base and Large tokenizers.

**Pretraining Hyperparameters**: For all models, we use a global batch size of 4096, and microbatch size of 128. We set the maximum sequence length during pretraining to 128, and we used the standard embedding dimension of 768. These hyperparameters were the same for MosaicBERT-Base and the baseline BERT-Base. More hyperparameter details are included in the Appendix.

## 3 Results

In our first set of experiments, we pretrained BERT-Base and MosaicBERT-Base for 70,000 steps of batch size 4096, which roughly corresponds to 78% of English C4 (Figure 1B and Figure 2). We ran experiments with $n = 5$ pretraining seeds for each model class. We then finetuned these models on the GLUE benchmark suite using identical finetuning parameters for all models and experiments; the details can be found in the Appendix.

### 3.1 MosaicBERT-Base Achieves 79.6 Average GLUE (dev) Score in 1.13 Hours

MosaicBERT-Base achieves the downstream average GLUE (dev) score of 79.6 in 1.13 hours on $8\times$A100 80 GB GPUs at a cost of roughly \$20 on a standard cloud provider. More details on cost estimates are included in the Appendix.

---

[10]The NVIDIA Apex library and Megatron [53] both use a form of low precision LayerNorm in their code, but it is not documented in any papers that we could find.

[11]Note that in the original BERT study, the vocabulary size was 30,000 [14]. However, the default vocab size for the `bert-base-uncased` tokenizer in HuggingFace is 30,522.

[12]With vocab size that is divisible by 64, the calculations go down a different kernel path with much higher occupancy.

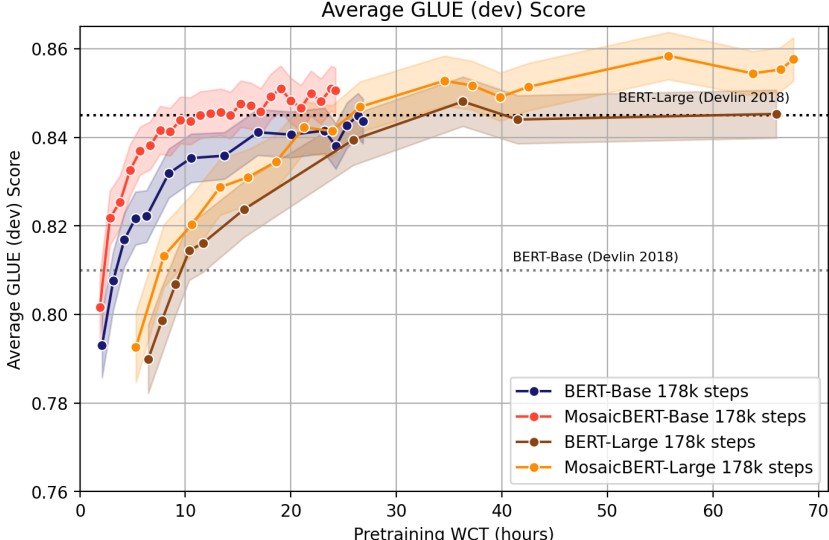

Figure 3: Average GLUE (dev) score Pareto curves for MosaicBERT-Base and Large trained for roughly 2 epochs of C4 (i.e. 178,000 steps with batch size 4096 with maximum sequence length 128 tokens). MosaicBERT-Base and Large are Pareto optimal relative to BERT-Base and Large. All pretraining is done on $8\times$A100 80GB devices (n=2-3 pretraining seeds). Note that BERT-Base and MosaicBERT-Base took much less time to train than BERT-Large and MosaicBERT-Large.

The baseline BERT-Base reached an average GLUE (dev) score of 83.2% in 11.5 hours (A100-80GB), while MosaicBERT reached the same accuracy in roughly 4.6 hours on the same hardware, which is roughly a $2.38\times$ speedup (Table 1 and Figure 1B).

### 3.2 MosaicBERT-Base is Pareto Optimal

As can be seen Figure 1B, MosaicBERT-Base consistently achieves higher average GLUE accuracy more quickly than the standard BERT-Base across all training durations. The performance of MosaicBERT on individual GLUE finetuning tasks can be seen in Figure 2. MosaicBERT-Base outperforms BERT-Base in four out of eight GLUE tasks across pretraining durations.

The space of NLP benchmarks and tasks has exploded in recent years; we include more information on the individual tasks in the classic GLUE benchmark in the Appendix. MNLI, QNLI and QQP are the largest datasets in the GLUE benchmark, with 100k-400k training examples, and MosaicBERT-Base is strictly Pareto-optimal for these tasks relative to BERT-Base (Figure 2). We interpret this to mean that the architectural changes and training recipe we chose resulted in an optimized, efficient BERT-Base model.

The quality of both models on smaller datasets (3k-67k training samples) is much more variable, as shown by the large error bars (standard deviation across n=5 pretraining seeds) for SST-2, MRPC and STSB. Regardless of this variation, MosaicBERT-Base performs equivalently to BERT-Base on these tasks across training duration.

### 3.3 MosaicBERT-Large is Pareto Optimal

While BERT-Base is one of the most popular BERT models, BERT-Large comes in a close second. All of our model development was done on MosaicBERT-Base; we were therefore curious whether our architecture and pretraining choices also generalized to a larger model.

In a second set of experiments, we pretrained MosaicBERT-Base and Large as well as BERT-Base and Large for two epochs of the C4 dataset. The training duration is 178,000 steps with batch size 4096, and is more than twice as long as the duration of the models in Figures 1 and 2.

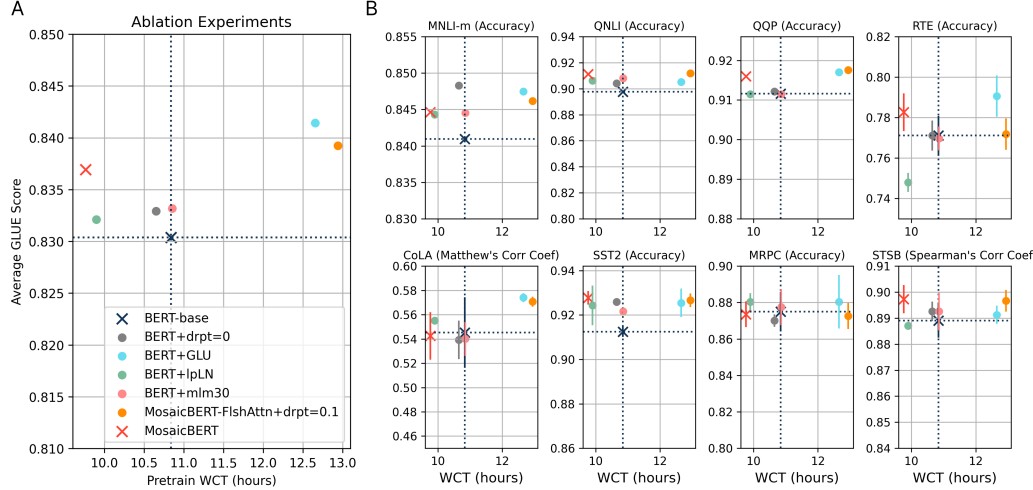

Figure 4: Ablation Experiments (A) Average GLUE score and (B) Individual GLUE tasks.
**BERT-base**: standard BERT-base (110M parameters) with attention dropout=0.1 and feedforward dropout=0.1, vocab size set to 30522, MLM=15% (all Hugging Face standard configurations). **BERT+drpt=0**: standard BERT-base, except that the attention in the dropout layer is set to 0 instead of the default 0.1. **BERT+GLU**: standard BERT-base, with GLU for the feedforward component of the encoder block. **BERT+lpLN**: standard BERT-base, except with low precision LayerNorm (bfloat16). **BERT+mlm30**: standard BERT-base, except with a masked language modeling masking ratio of 30%. **MosaicBERT**: the complete MosaicBERT-Base including GLU (where the dimension of the intermediate layer is 3072 resulting in 137M total parameters), ALiBi, low precision LayerNorm, unpadding, MLM 30%, vocab size 30528 (a multiple of 64) and the attention dropout=0. **MosaicBERT-FlashAttn+drpt=0.1**: MosaicBERT-Base *without* Flash Attention and *with* the attention dropout set to 0.1.

BERT-Large has 24 repeated transformer layers (BERT-Base has 12), and as a result consumes much more memory and takes longer to train. We found that MosaicBERT-Large reached an average GLUE score of 83.2 in 15.85 hours, while BERT-Large took 23.35 hours (Figure 3). MosaicBERT-Large therefore had a $1.47\times$ speedup over BERT-Large in this training regime.

While MosaicBERT-Base is optimal for a constrained budget, the MosaicBERT-Large average GLUE score eventually surpasses MosaicBERT-Base after 25 hours of training on a $8\times$A100-80GB node, and reaches an average score of 85.5 in roughly 50 hours. In our experiments, the MosaicBERT-Large architecture and pretraining was the same as MosaicBERT-Base, outside of the number of attention heads, number of hidden layers, and intermediate size of the feedforward units.

A striking result from Figures 3 and S2 is that MosaicBERT-Base is Pareto optimal relative to BERT-large in this training regime, and is Pareto optimal to MosaicBERT-Large during the first half of training. MosaicBERT-Base takes only 25 hours to complete 2 epochs, while MosaicBERT-Large takes close to 70 to complete 2 epochs. A potential takeaway from this is that MosaicBERT-Large only surpasses Base performance in the large data regime. For certain tasks such as QQP, SST-2, and MRPC MosaicBERT-Base achieves a maximum accuracy on par with the maximum accuracy of MosaicBERT-Large, for far fewer pretraining hours. When building encoders for specific domains and tasks, bigger is not always better.

### 3.4 Ablation Experiments

When deciding on the final setup for MosaicBERT, we tried to make educated guesses about what combination of modifications could work well together to both increase FLOP utilization and maintain or improve accuracy. Our approach was inspired by Amdahl's Law, i.e. the idea that optimizing a portion of a system responsible for N% of the costs can lead to, at most, an N% speedup [48], and chose each of the modifications such that they did not target the same exact part of the of the stack.

For example, low precision LayerNorm and GLU affect different parts of the encoder block, and so potentially compose well.

In the following ablation experiments we benchmark the individual effects of various architecture and training choices on pretraining wall clock time. We plot downstream GLUE accuracy as a function of measured pretraining wall clock time in Figure 4.

The patterns in Figure 4A-B shed light on the individual effects of various architectures (e.g. BERT+GLU, BERT+low precision LayerNorm) and training configurations (e.g. BERT + 30% masking ratio). On average, all methods seem to provide a slight accuracy boost to BERT-base. Increasing the masking ratio to 30% leads to a slight accuracy boost while not affecting the WCT, while turning off dropout in the attention layer (BERT+drpt=0) leads to a slight improvement in both accuracy and WCT. Low precision LayerNorm (BERT+lpLN) leads to a significant speedup (i.e. a shift to the left). Gated Linear Units (BERT+GLU) add more parameters to the model and lead to a significant slowdown while providing an accuracy boost. As a point of reference, we also benchmark the full MosaicBERT as well as MosaicBERT without FlashAttention and with attention dropout set to 0.1 (the standard BERT-base configuration).

In these experiments, all BERT-Base models here were pretrained on C4 for 70,000 steps with batch size 4096, and microbatch size 256 on 8×A100 80GB GPUs. All models were initialized with the same seed and shared all other hyperparameters including the bert-base uncased tokenizer, the learning rate schedule, AdamW as the optimizer, etc. Final pretraining checkpoints were then finetuned on GLUE following the details in the appendix. The points represented in these GLUE plots are final finetuning checkpoints.

These plots highlight the importance of benchmarking with Pareto curves, as it is not possible to tell from these plots alone whether training BERT-base for 2 more hours leads to better performance than BERT+GLU, for example.

# 4   Related Work

Various studies rightly focus on a single method or modification that improves throughput, such as FlashAttention [11] and unpadding [66]. In this study, we incorporate many of these techniques to investigate whether they combine advantageously.

RoBERTa ("Robustly optimized BERT approach") is the most influential work in this regard [37]. In this study, they kept the exact BERT architecture but changed the training recipe by removing the next sentence prediction objective, training for longer on much larger datasets, and changing the batch size, among other things. They showed that the original BERT was significantly undertrained - while the original BERT trained on 16GB worth of text, the top accuracy RoBERTa (Large) was trained on 160GB of data for 500,000 steps with batch size 8192. Many of the training choices in RoBERTa have become standard practice; our training recipe therefore more closely resembles RoBERTa than the original BERT. See Appendix for a more detailed comparison of RoBERTa and MosaicBERT.

Improvements in transformer architecture and GPU hardware have caused the cost of pretraining BERT models to decline precipitously. The recent paper "Cramming: Training a Language Model on a Single GPU in One Day" [18] exemplifies this trend. The goal of this rigorous study was to train the best BERT in 24 hours on a single GPU. Similar to us, they tweaked the BERT architecture to incorporate FlashAttention and Gated Linear Units (but without increasing the dimensionality of the hidden block). Unlike MosaicBERT, they used scaled sinusoidal positional embeddings [59] as well as Pre-LayerNorm (applying LayerNorm before the attention and feedforward layers) and did not change the pretraining masking rate. With this setup, they were able to train their modified BERT-Base to an average GLUE score of 80.4 in 24 hours on a single A6000 GPU (i.e. 24 GPU hours, as per Table 3 of [18]). Our study is similar in spirit, but asks what is the fastest architecture for pretraining, and expands on this in greater detail by showing that MosaicBERT is Pareto optimal relative to BERT.

While we believe that our training optimization choices for MosaicBERT go a long way to improve BERT training efficiency, there are likely further modifications that could lead to an increase in accuracy with no effect on throughput, such as replacing LayerNorm with RMSNorm [67] and GeGLU with SwiGLU [50, 42]. Removing all biases from the linear layers can also potentially lead to slight speed ups in training time without a significant hit to accuracy. "NarrowBERT" [36]

suggested a clever change to the way encoder blocks are stacked so that computation is not wasted on unmasked tokens. Another recent study showed that dynamically masking the masking ratio during pretraining leads to downstream accuracy gains in BERT [1]. Note, however, that accounts such as [18] and [29] document many examples where certain modifications to the BERT architecture and training recipe fail to improve accuracy. We also note here that there is an ongoing debate about the benefits of RoPE (rotary positional embeddings) [55] versus AliBi [44]. Finally, using a more modern tokenizer will likely have an effect on downstream accuracy [18].

Approaches such as knowledge distillation [5] might additionally push the Pareto frontier of BERT models during pretraining and finetuning. Progressive layer stacking is one example technique; by initializing a larger BERT model with the weights from smaller pretrained BERT models, pretraining can be reliably accelerated [21, 29]. As the field is learning how to optimize stable model training at the billion parameter scale [8, 13], we expect some of these innovations to cycle back to smaller models such as BERT-Base. For example, it has been hypothesized that incorporating more Layer-Norm modules at the QK matrix output (i.e. QK-LayerNorm [24]) can lead to improved stability during training; combining this with a more aggressive learning rate schedule could lead to faster convergence.

## 5 Conclusion

In this study we show how combinations of architecture choices can improve BERT pretraining speed and accuracy. We built MosaicBERT to enable ML researchers and engineers to pretrain BERT models from scratch on their own data and build better models for their specific domains without facing time and cost restrictions. We ultimately hope that by making BERT training faster and cheaper, our work contributes to the trend in NLP research away from finetuning generic models and towards of pretraining custom encoders on domain specific data. A large body of research has highlighted the success of BERT-style models pretrained on a specific domains such as biomedicine [3, 33, 22, 17], math [51], chemistry [2, 25], financial communications [49], and code [56]. This seems to hold true even in the age of LLMs [6, 65].

**Code**

A stable webpage for this work can be found at `mosaicbert.github.io`. Code for pretraining and finetuning MosaicBERT can be found in the MosaicML `examples` repository `https://github.com/mosaicml/examples`. The exact code for this study was pinned to `v0.0.4` of the MosaicML `mosaicml/examples` repository `https://github.com/mosaicml/examples/tree/v0.0.4/examples/bert`. All pretraining and finetuning was done in PyTorch 1.13 using the MosaicML Composer library `https://github.com/mosaicml/composer`. Model weights for MosaicBERT-Base can be found on the HuggingFace hub `https://huggingface.co/mosaicml/mosaic-bert-base`.

Code for pretraining and finetuning transformers such as MPT and derivative versions of MosaicBERT can be found in the MosaicML `llm-foundry` repository `https://github.com/mosaicml/llm-foundry`.

**Acknowledgements**

The authors would like to thank Erica Yuen, Vitaliy Chiley, Connor Jennings, and Aaron Gokaslan for their feedback on the code and manuscript. We would also like to thank Ishana Shastri and Vlad Ivanchuk for some of their early work as interns that led to MosaicBERT. The majority of this work appeared as a blogpost with the title "MosaicBERT: Pretraining BERT from Scratch for \$20" in March 2023.[13] This work also builds on the MosaicML MLPerf v2.1 results for BERT-Large [31].

---

[13]`mosaicml.com/blog/mosaicbert`

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

# A Pretraining Hyperparameters

We detail our pretraining hyperparameter values in Table S1 below.

| Model | Optimizer | LR | beta | eps | WD | MB | Warmup | Final LR | MLM |
|---|---|---|---|---|---|---|---|---|---|
| BERT-Base | Decoupled AdamW | 5e-4 | [0.9,0.98] | 1e-6 | 1e-5 | 512 | 6% | 0.02LR | 0.15 |
| MosaicBERT-Base | Decoupled AdamW | 5e-4 | [0.9,0.98] | 1e-6 | 1e-5 | 512 | 6% | 0.02LR | 0.3 |
| BERT-Large | Decoupled AdamW | 2e-4 | [0.9,0.98] | 1e-6 | 1e-5 | 256 | 6% | 0.02LR | 0.15 |
| MosaicBERT-Large | Decoupled AdamW | 2e-4 | [0.9,0.98] | 1e-6 | 1e-5 | 256 | 6% | 0.02LR | 0.3 |

Table S1: Pretraining hyperparameters for BERT-Base, MosaicBERT-Base, BERT-Large and MosaicBERT-Large. LR = learning rate, WD = weight decay, MB = Microbatch and MLM = Masked Language Modeling ratio. All pretraining runs were done on $8\times$ A100-80 GB GPUs.

All pretraining runs followed a warmup + linear decay learning rate schedule. Evaluation was done every 2000 batches, and checkpoints were saved every 3500 batches - both of which slow down training slightly.

# B Finetuning Hyperparameters

We used the hyperparameters in Table S2 for finetuning all BERT and MosaicBERT models. All finetuning datasets used a maximum sequence length of 256 tokens. We found that these values worked well across all tasks for BERT-Base, MosaicBERT-Base, and MosaicBERT-Large; BERT-Large however was somewhat less performant on QQP for some pretraining seeds.

| Task | learning rate | beta | epsilon | weight decay | epochs | seeds |
|---|---|---|---|---|---|---|
| MNLI | 5e-5 | [0.9, 0.98] | 1e-6 | 5e-6 | 3 | 1 |
| QNLI | 1e-5 | [0.9, 0.98] | 1e-6 | 1e-6 | 10 | 1 |
| QQP | 3e-5,1.5e-5 | [0.9,0.988] | 1e-6 | 3e-6 | 5 | 1 |
| RTE | 1e-5 | [0.9, 0.98] | 1e-6 | 1e-5 | 3 | 5 |
| CoLA | 5e-5 | [0.9, 0.98] | 1e-6 | 5e-6 | 10 | 4 |
| SST-2 | 3e-5 | [0.9,0.988] | 1e-6 | 3e-6 | 3 | 3 |
| MRPC | 8e-5 | [0.9, 0.98] | 1e-6 | 8e-6 | 10 | 5 |
| STS | 3e-5 | [0.9, 0.98] | 1e-6 | 3e-6 | 10 | 5 |

Table S2: Finetuning hyperparameters for BERT and MosaicBERT across Base and Large. All finetuning was done with sequence length 256. Note that finetuning BERT-Large on QQP with learning rate 3e-5 was highly unstable; we therefore lowered the learning rate in this instance to 1.5e-5 for BERT-Large (but not MosaicBERT-Large).

As can be seen from Figure S2, we found the finetuning of BERT-Large to be highly variable. Finetuning runs that did not converge, or that led to outlier final values were dropped. For this reason, not all checkpoints are represented in the BERT-Large average GLUE plot in Figure 3.

We did not do a careful hyperparameter sweep for BERT-Large finetuning, and instead tried to keep values consistent across BERT-Base and Large as well as MosaicBERT-Base and Large.

# C Pareto Curves for Benchmarking Neural Network Training

In order to make meaningful improvements in neural network training efficiency, ML practitioners must be able to compare between different choices of network architectures, hyperparameters, and training algorithms. One straightforward way to do this is to characterize the tradeoff between

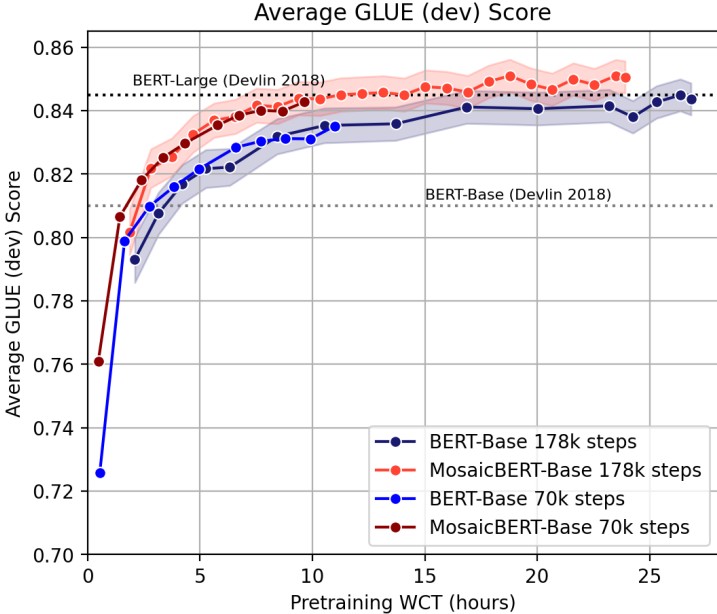

Figure S1: Pareto curves for BERT-Base and MosaicBERT-Base for runs trained for 70,000 and 178,000 steps with batch size 4096 and sequence length 128. Same data as Figure 1B and 3.

accuracy and training time with a "tradeoff curve" or a "Pareto curve." Pareto curves can be generated by varying the length of training for each model configuration; longer training runs take more time but tend to reach higher quality. For a fixed model and task configuration, this method of generating tradeoff curves is an estimate of the theoretical Pareto frontier, i.e. the set of all of the best possible tradeoffs between training time and accuracy, where any further attempt to improve one of these metrics worsens the other. See [40, 43] for more details.

We therefore consider one model Pareto optimal relative to another if it has superior accuracy across different training budgets while keeping everything fixed. Many studies advertise novel architecture approaches without measuring wall clock time, or simply show an improvement for a single training duration. In this study we show that MosaicBERT-Base is Pareto optimal relative to our BERT-Base baseline for both short training durations and long training durations (Figures 1, 2, 3). We also show that BERT-Large and MosaicBERT-Large are not Pareto optimal relative to BERT-Base and MosaicBERT-Base for training durations under 30 hours (Figure 3).

The Pareto plots in Figures 1, 2, 3 and S2 are constructed by taking points from the same runs (with the same learning schedule), instead of having one run per pretraining budget on the plot. We did this for cost reasons; it would have been too expensive to do every run separately.

For ResNets, it has been established that the most accurate way to generate Pareto curves is to run completely separate training runs for each point on the Pareto curve, where each point also has their own learning rate schedule [43]. However, to the best of our knowledge, this has not been clearly established for BERT models. Can Pareto curves for BERT simply be constructed by evaluating checkpoints of a single training run?

Since we ran BERT-Base and MosaicBERT-Base experiments for two separate learning rate schedules (70,000 steps and 178,000 steps with batch size 4096), we can plot the Pareto curves for all of these runs on top of each other (Figure S1). Somewhat surprisingly, the 70k and 178k curves for BERT-Base (and MosaicBERT-Base) mostly lie on top of each other. This is strong evidence that at least for BERT models, a reasonable Pareto curve can be constructed using checkpoints from a single training run.

We note that the Pareto curve approach discussed here is also related to "tokens per parameter" (TPP) considerations that are often discussed in LLM pretraining (e.g. see [15]).

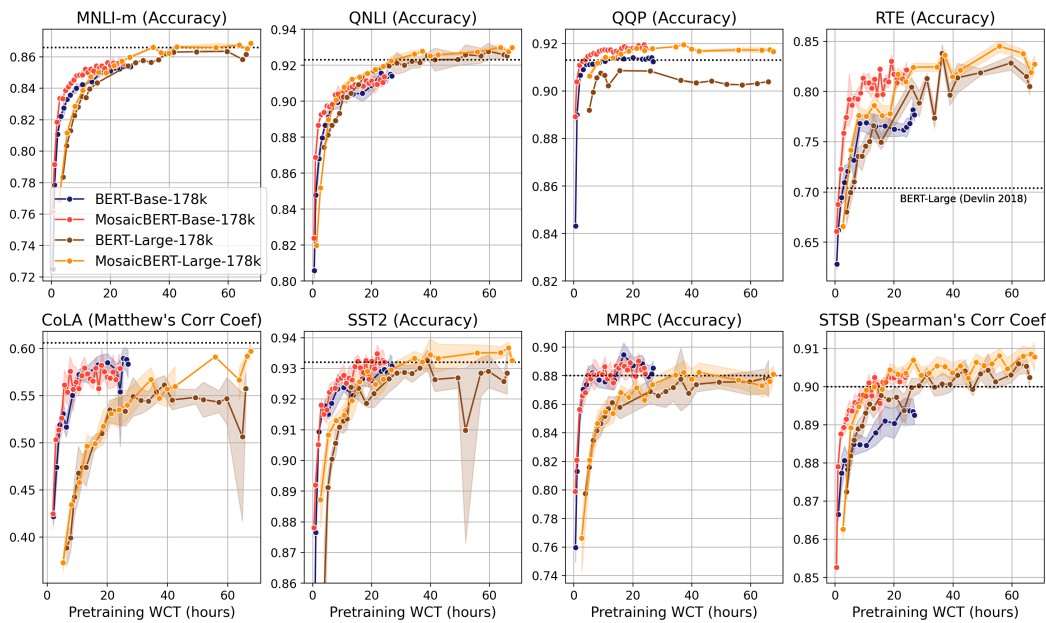

Figure S2: MosaicBERT-Base and Large accuracy (dev) vs. pretraining speed Pareto curves for individual GLUE benchmarks. All models were trained on roughly 2 epochs of C4 (178,000 steps with batch size 4096 with maximum sequence length of 128). Error bars represent 95% confidence interval for n=2-3 pretraining seeds. Dashed black line represents BERT-Large accuracy on the GLUE (dev) data as reported in [37]. The average GLUE score can be found in Figure 3.

## D MosaicBERT-Base Experiments with Increasing Sequence Length

Batch size, maximum sequence length, and number of training steps are important hyperparameters that affect both downstream GLUE accuracy as well as wall clock time. While increasing the maximum number of tokens in a sequence leads to a direct increase in wall clock time, it also exposes the model to more data. For example, increasing sequence length from 128 to 256 while keeping all other hyperparameters constant leads to $2\times$ the amount of total training tokens (assuming no padding). However, this will lead to a commensurate increase in pretraining wall clock time. Given our Pareto-based approach, we can therefore ask: *is it better to train with fewer steps and longer sequence length, or more steps and shorter sequence length?*

Here we find that training for more steps (i.e. 178,000) on shorter sequence length (i.e. 128 tokens) is Pareto optimal for GLUE scores.

As part of our experimental benchmarking, we trained MosaicBERT-Base models with varying maximum sequence lengths ranging from 256 tokens to 2048 tokens for 70,000 steps with batch size 4096. We then plotted the downstream GLUE accuracy in Figures S3 and S4. Besides maximum sequence length, all other pretraining and finetuning hypeparameters are the same as MosaicBERT-Base in the main text (and Tables S1 and S2). Each point represents n=1-2 pretraining checkpoints finetuned with multiple seeds according to Table S2. The curve representing MosaicBERT-Base trained with sequence length 128 and 178,000 steps is the same as in Figure 3.

When looking at average GLUE accuracy as a function of pretraining steps in Figure S3A, longer sequence lengths of 256, 512, 1024 and 2048 nominally do better than a maximum sequence length of 128 tokens when trained for 70,000 steps. However, plotting steps on the x-axis can be misleading, as increasing sequence length leads to increased wall clock time (ceteris paribus).

Instead, it is much more informative to plot wall clock time on the x-axis, as in Figure S3B. Here, the order of the curves is flipped and the curve representing maximum sequence length of 128 tokens is mostly Pareto optimal (red curve). Interestingly, training with batch size and sequence length of {4096,128} for 178,000 steps leads to the same final average GLUE accuracy *and wall clock time* as training with {4096, 512} for 70,000 steps (purple curve in Figure S3B). Note that the x-axis here is

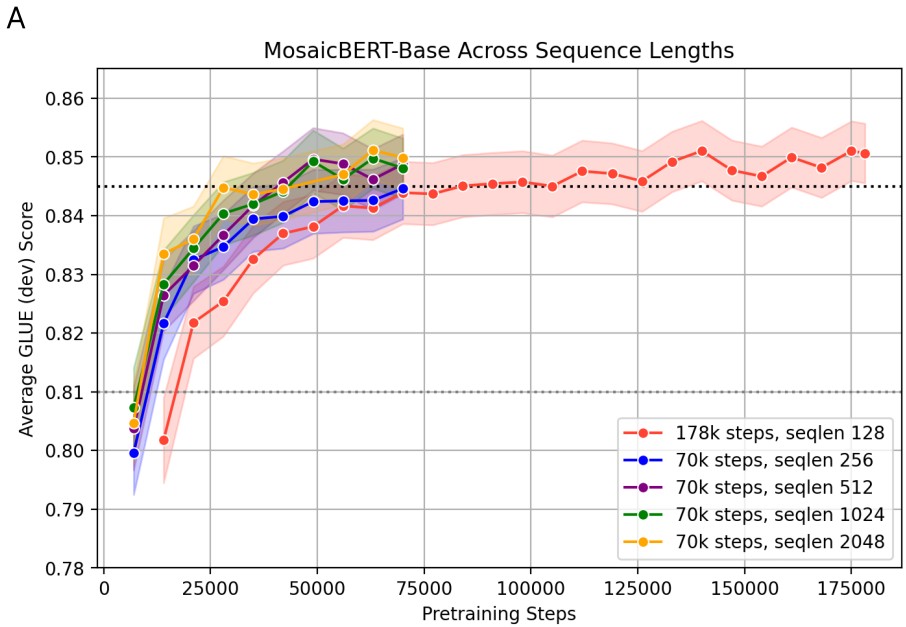

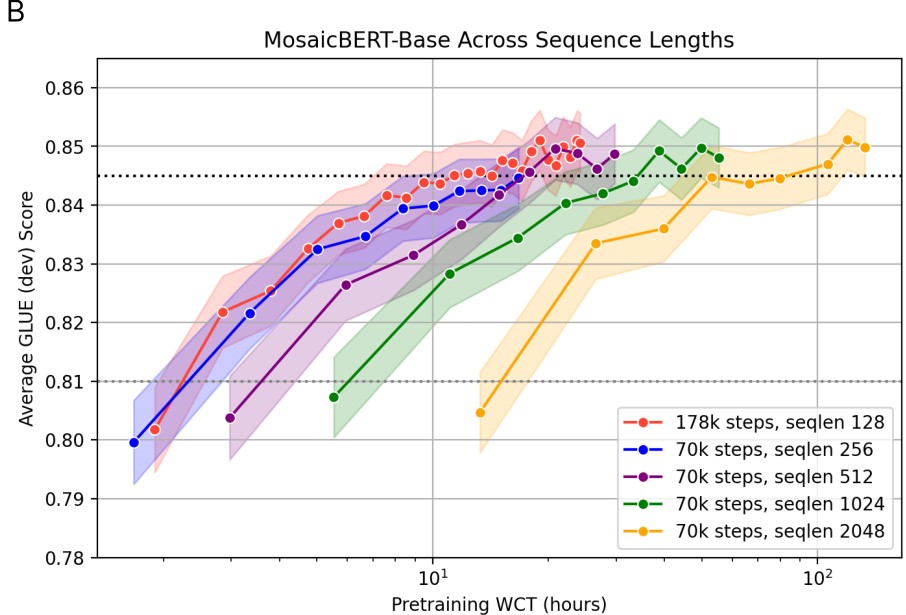

Figure S3: MosaicBERT-Base Pareto curves for maximum sequence lengths ranging from 128 to 2048 tokens, with all other hyperparameters fixed. (A) Average GLUE score as a function of pretraining steps. (B) Same data as in A, but with wall clock time on the x-axis. The curve representing a maximum sequence length of 128 tokens is mostly Pareto optimal (red curve). Interestingly, training with batch size and sequence length of {4096,128} for 178,000 steps (red curve) leads to the same final average GLUE accuracy *and wall clock time* as training with {4096, 512} for 70,000 steps (purple curve). Top dashed line represents BERT-Large average (dev) GLUE score from Devlin et al. 2018, while bottom dashed line represents the BERT-Base average (dev) GLUE score from Devlin et al. 2018 [14].

plotted on a logarithmic scale, as training with sequence length 2048 for 70,000 steps takes more than 100 hours on 8×A100 80GB GPUs.

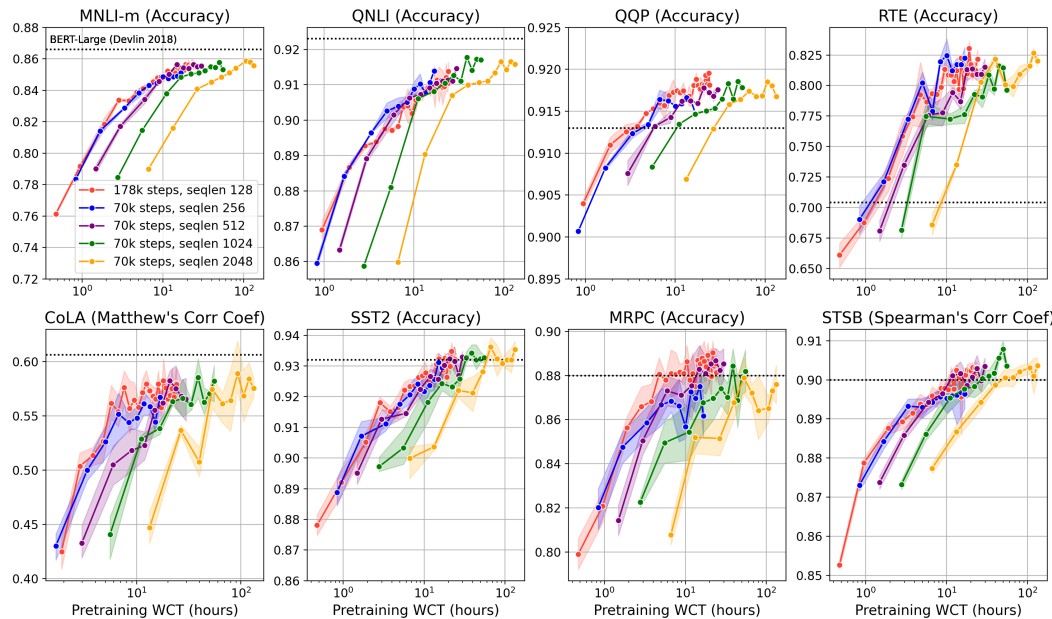

Figure S4: MosaicBERT-Base Pareto curves for maximum sequence lengths ranging from 128 to 2048 tokens, with all other hyperparameters fixed. Individual GLUE scores are plotted as a function of pretraining wall clock time. All curves used batch size 4096 and were pretrained for 70,000 steps, except for the curve representing sequence length 128 tokens, which was trained for 178,000 steps (and is the same as MosaicBERT-Base in Figure 3).

Figure S4 plots the GLUE scores for individual tasks as a function of pretraining wall clock time. The curve representing maximum sequence length of 128 tokens is mostly Pareto optimal (red curve). However, QNLI and STSB seem to benefit from longer maximum sequence lengths.

An important consideration when increasing the maximum sequence length is the issue of padding. Many text samples in our C4 pretraining dataset contained fewer than 2048 tokens and needed to be padded. The "unpadding" approach detailed in the main text applies to the feedforward layers of the transformer but not the attention layers. As sequence length is increased, there is potentially wasted computation on padding tokens in the attention layers. This means that wall clock time increases without significant boosts in accuracy. This is likely the case for the curves representing sequence lengths 1024 and 2048.

Another important consideration is that the GLUE tasks are quite short by modern standards (circa 2023) and contain fewer than 512 tokens. Additionally, all GLUE finetuning in this study was done with maximum sequence length of 256 tokens. Training with sequence length 1024 or 2048 tokens is likely unnecessary for this task. It is possible, however, that these models are superior at tasks that involve long-context dependencies.

The original BERT paper pretrained with a maximum sequence length of 128 tokens for 90% of training, and then completed pretraining with a sequence length of 512 [14]. The RoBERTa paper pretrained with a maximum sequence length of 512 for the full duration of pretraining and also "packed" each batch sample with full sentences up to the limit of 512 tokens to decrease the number of padding tokens [37].

We released the pretrained checkpoints for these models on the HuggingFace hub in April 2023:

- `https://huggingface.co/mosaicml/mosaic-bert-base-seqlen-256`

- `https://huggingface.co/mosaicml/mosaic-bert-base-seqlen-512`

- `https://huggingface.co/mosaicml/mosaic-bert-base-seqlen-1024`

- `https://huggingface.co/mosaicml/mosaic-bert-base-seqlen-2048`

| Model | Bsz | Steps | SeqL | MNLI | QNLI | QQP | RTE | SST | MRPC | CoLA | STS | Av. |
|---|---|---|---|---|---|---|---|---|---|---|---|---|
| **BERT-Large** BookC + Wiki | 256 | 1M | 128 | 86.6 | 92.3 | 91.3 | 70.4 | 93.2 | 88 | 60.6 | 90 | 84.05 |
| **RoBERTa-Base** all data+500k | 8k | 500k | 512 | 87.6 | 92.8 | 91.9 | 78.7 | 94.8 | 90.2 | 63.6 | 91.2 | 86.35 |
| **RoBERTa-L**: | | | | | | | | | | | | |
| 1. BookC+Wiki | 8k | 100k | 512 | 89 | 93.9 | 91.9 | 84.5 | 95.3 | 90.2 | 66.3 | 91.6 | 87.8 |
| 2. all data | 8k | 100k | 512 | 89.3 | 94 | 92 | 82.7 | 95.6 | 91.4 | 66.1 | 92.2 | 87.9 |
| 3. all data+300k | 8k | 300k | 512 | 90 | 94.5 | 92.2 | 83.3 | 96.1 | 91.1 | 67.4 | 92.3 | 88.4 |
| 4. all data+500k | 8k | 500k | 512 | 90.2 | 94.7 | 92.2 | 86.6 | 96.4 | 90.9 | 68 | 92.4 | 88.9 |
| **BERT-Base** | 4k | 178k | 128 | 85.4 | 91.6 | 91.4 | 78.2 | 93.10 | 89.5 | 59 | 89.4 | 84.7 |
| **MosaicBERT-B** | 4k | 178k | 128 | 85.6 | 91.4 | 92 | 83 | 93.5 | 89 | 58.2 | 90.3 | 85.4 |
| **BERT-Large** | 4k | 178k | 128 | 86.3 | 92.8 | 90.9 | 83.8 | 93.3 | 87.8 | 56.2 | 90.6 | 85.2 |
| **MosaicBERT-L** | 4k | 178k | 128 | 86.9 | 93 | 92 | 84.5 | 93.7 | 88.2 | 59.7 | 90.9 | 86.1 |

Table S3: Model performance comparison on GLUE (dev) tasks. Data from Table 8 of the RoBERTa paper [37]. Values for BERT and MosaicBERT in bottom rows are best average values per task from Figure S2.

## E   Comparison to RoBERTa

As mentioned in the main text, RoBERTa ("Robustly optimized BERT approach") was an important follow-up to the original BERT [37]. In this study, they kept the exact BERT architecture the same but changed the *training recipe* by removing the next sentence prediction (NSP) objective, training for longer on much larger datasets, and changing the batch size, among other things. Many of the training choices in RoBERTa have become standard practice; our training recipe therefore more closely resembles RoBERTa than the original BERT. The RoBERTa paper also did careful experiments and ablations that were helpful for the research community.

The main pretraining results from RoBERTa-Base and Large are shown in Table D (data from Table 8 in the Appendix of the RoBERTa paper [37]). In particular, they trained on:

1. BookCorpus [68] and English Wikipedia for 100k steps with batch size 8192 and sequence length 512

2. BookCorpus [68] and English Wikipedia (16GB), plus "additional data" CC News, the English portion of the CommonCrawl News dataset (76GB) [41], OpenWebText Corpus (38GB) [20], and STORIES (31GB) [58] for 100k steps with batch size 8192 and sequence length 512

3. The full dataset mixture (160GB) for 300k steps with batch size 8192 and sequence length 512

4. The full dataset mixture (160GB) for 500k steps with batch size 8192 and sequence length 512

The RoBERTa-Base and Large values are better than the MosaicBERT-Base and Large values. An apples-to-apples comparison of RoBERTa and MosaicBERT experiments would keep all hyperparameters consistent, including training data and batch size. Since we cannot do this post hoc, a somewhat reasonable comparison would be to keep the total amount of data constant. 100k steps with batch size 8192 is roughly equivalent to 178k steps with batch size 4096 (i.e. it is almost equivalent to 89k steps with batch size 8192). However, RoBERTa also pretrained with a maximum sequence length of 512 tokens, *which is $4\times$ longer than the maximum sequence length of 128 we used for pretraining in this study*. This means that the "worst" RoBERTa models trained with batch size 8192 for 100k steps still saw $4\times$ more tokens than the MosaicBERT models trained with batch size 4096 for 178k steps. This is one likely one reason that the final RoBERTa values on GLUE are higher.

Additionally, the RoBERTa study formatted the data to minimize padding tokens by packing each sample with full sentences sampled contiguously from one or more documents (with an extra separator token) [37]. They also did a learning rate sweep for single-task GLUE finetuning {1e-5, 2e-5, 3e-5}, which we did not do. Finally, the differences between MosaicBERT and RoBERTa could also be due to differences in the underlying datasets.

# F    GLUE Benchmark Details

The GLUE benchmark consists of 8 (originally 9) tasks [60]. Since there has been a Cambrian explosion of benchmarks since the halcyon days of GLUE, we elaborate on the individual GLUE benchmarks for reference. We used the GLUE tasks from HuggingFace for finetuning and evaluation: `https://huggingface.co/datasets/glue`. All evaluation was done on the validation (a.k.a. dev) splits.

## F.1    Large Finetuning Datasets

**MNLI (Multi-Genre Natural Language Inference)** [392,702 train | 19,643 validation | 19,643 test] is a large crowd-sourced entailment classification task [64]. The model is given two sentences and has to predict whether the second sentence is entailed by, contradicts, or is neutral with respect to the first one. For example:

- Premise: "Buffet and a la carte available."

- Hypothesis: "It has a buffet."

- Label: 0 (entailment)

MNLI has two subsets, "matched" (MNLI-m) and "mismatched" (MNLI-mm). All numbers reported in this study are for MNLI-m, unless otherwise stated.

**QNLI** [104,743 train | 5,463 validation | 5,463 test] this Stanford Question Answering dataset consists of question-paragraph pairs drawn from Wikipedia [47].

**QQP (Quora Question Pairs 2)** [363,846 train | 40,400 validation | 390,965 test]. The task is to determine whether two sentences are semantically equivalent [26].

## F.2    Small Finetuning Datasets

**RTE (Recognizing Textual Entailment)** [2,490 train | 277 validation | 3,000 test] Given two sentences, the model has to predict whether the second sentence is or is not entailed by the first sentence [10, 19, 4]. Note that in our work we use a checkpoint from the MNLI finetuning to finetune on RTE.

**CoLA (Corpus of Linguistic Acceptability)** [8,551 train | 1,040 validation | 1,063 test] [62] is a benchmark with sentences that are either linguistically acceptable or grammatically incorrect. For example:

- "The higher the stakes, the lower his expectations are." Label: 1 (acceptable)

- "Mickey looked up it." Label: 0 (unacceptable)

**SST-2 (Stanford Sentiment Treebank)** [67,349 train | 872 validation | 1,821 test] consists of sentences from movie reviews. The task is to classify the sentiment as either positive or negative [54].

**MRPC (Microsoft Research Paraphrase Corpus)**[3,668 train | 408 validation | 1,725 test] [16] The dataset consists of sentence pairs extracted from online news sources. The task is to classify whether the sentences in the pair are semantically equivalent.

**STSB (Semantic Textual Similarity Benchmark)** [5,749 train | 1,500 validation | 1,379 test] This dataset contains sentence pairs that are given similarity scores from 0 to 5 [7].

Note that we excluded finetuning on the 9th GLUE task WNLI (Winograd NLI) [34], as in the original BERT study (it is a very small dataset [634 train, 146 test] with a high number of adversarial examples). Finetuning on RTE, MRPC and STSB starts from a checkpoint already finetuned on MNLI (following the example of [28] and other studies). This is done because all the above tasks deal with sentence pairs, and this staged finetuning leads to consistent empirical improvement.

# G  MosaicBERT-Large Multinode Throughput Scaling

The experiments in the main section of this paper were all performed on a single node with $8\times$A100 GPUs. How well do our innovations to the BERT architecture maximize throughput at the multinode scale?

We measured the throughput of MosaicBERT-Large (430M) during training on 8, 16, 32, 64, 128 and 200 GPUs, and plotted the tokens per second for various global batch sizes. Global batch size is an important factor in the throughput measurements; in general, cranking up the batch size increases the GPU utilization and raw throughput. As the number of nodes increases, the global batch size needs to be increased as well in order to maintain high throughput.

If the global batch size is kept constant while increasing the number of nodes, the throughput does not increase linearly. This can be seen in Figure S5; a global batch size of 4096 spread across 64 GPUs using Distributed Data Parallelism (DDP) means that each GPU will only apply matmul operations on matrices with a dimension of 64, which leads to suboptimal throughput. If the global batch size is increased to 65,536 across 64 GPUs, this roughly means that each GPU will apply matmul operations on matrices with a dimension of 1024, leading to higher throughput. However, such a large global batch size might not lead to the best downstream accuracy; this is a question that we were not able to address in this study due to resource and time constraints.

## G.1  How MosaicBERT modifications might scale to larger models

All of the architectural and configuration choices made in MosaicBERT can in principle be applied to larger models, although they might have different effects on throughput. Additionally, they should not pose any issues with different forms of parallelism (some of the modifications we explored were used successfully at large scale in frameworks like MEGATRON as well as MPT-7B and MPT-30B [39, 38]). For example, Gated Linear Units should have no issues with tensor parallelism as long as the matrix multiplications are a multiple of the Tensor Parallel world size. As models are scaled, LayerNorm becomes a smaller portion of the total compute, so low precision LayerNorm might not matter as much.

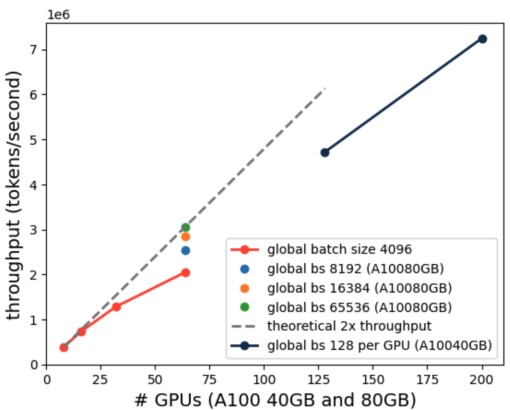

Figure S5: MosaicBERT-Large (430M) multinode throughput scaling

# H  MosaicBERT-Base Model FLOPs Utilization (MFU)

Model FLOPs Utilization (MFU) is an estimate of what percentage of the hardware's FLOPs are being used during training. The estimate is based on the measured throughput and the known FLOPs of the computation.

MFU calculates the utilization from the floating point operations required for a single forward/backwards pass of the model, and does not account for the additional compute required for

| Model | Throughput (tokens /sec) | MFU | Hardware | Time to 79.6 | Batch Size | Micro-batch Size |
|---|---|---|---|---|---|---|
| BERT Base | 0.4e6 | 10.4% | 8× A100 80 | 110.4 minutes (1.84 hours) | 4096 | 512 |
| MosaicBERT Base | 1.1e6 | 39.97% | 8× A100 80 | 67.8 minutes (1.13 hours) | 4096 | 512 |
| MosaicBERT Base | 0.938e6 | 30.9% | 8× A100 40 | 76.8 minutes | 4096 | 128 |
| MosaicBERT Base | 1.88e6 | 31.0% | 16× A100 40 | 38.5 minutes | 4096 | 128 |
| MosaicBERT Base | 3.15e6 | 25.9% | 32× A100 40 | 23.1 minutes | 4096 | 128 |
| MosaicBERT Base | 4.77e6 | 19.6% | 64× A100 40 | 15.7 minutes | 4096 | 64 |

Table S4: Multinode Throughput scaling for MosaicBERT-Base

other implementation details such as activation checkpointing. Thus, MFU is independent of implementation and hardware. For more details, see [32]. All FLOP calculations exclude the operations required for normalization, activation, and residuals.

Following the notation in the PaLM paper [8], Model FLOPs Utilization (MFU) is approximated as:

$$\text{MFU} = \frac{(6 \cdot n_{parameters} \cdot T_{observed})}{n_{gpus} \cdot T_{theoretical}} \tag{3}$$

where $T_{observed}$ is the observed throughput and $T_{theoretical}$ is the theoretical peak throughput.

In the numerator, the number of learnable parameters in the model is multiplied by a factor of 6 to estimate the matmul FLOPs per token seen ($2\times$ for the forward pass and $4\times$ for the backward pass). This is then multiplied by the number of tokens seen per second. As a first-order approximation, we exclude the extra FLOPs per token due to dense self-attention.

In the denominator, the theoretical peak throughput is provided in the GPU hardware specs. For A100 GPUs using `bfloat16`, this theoretical peak throughput is 312 teraFLOPs.

| MosaicBERT-Base Ave. GLUE Score | 8×A100 80GB hours | 8×A100 80GB cost ($2.50 GPU/hr) | 8×A100 40GB hours | 8×A100 40GB cost ($2 GPU/hr) |
|---|---|---|---|---|
| 79.6 | 1.13 | $22.60 | 1.28 | $20.00 |
| 82.2 | 2.81 | $56.20 | 3.19 | $51.00 |
| 83.4 | 5.27 | $105.40 | 5.99 | $95.78 |

Table S5: MosaicBERT-Base GLUE (dev) scores, time and cost comparison

# I  GPU Pricing

As of mid-2023, A100 GPU pricing ranges from $4.10 (40 GB) for on demand cloud compute on AWS, to $2.46 (40 GB) / $5.00 (80 GB) per GPU on GCP to $1.10 (40 GB) / $1.50 (80 GB) per GPU using Lambda labs. At an intermediate price of $2.50 an hour per A100 80 GB GPU, training to 79.6 GLUE average score takes 1.13 hours and costs roughly $22.60.[14] Some example costs are calculated in Table S5.

---

[14]See for example "Cloud GPU instances with the largest VRAM 2022" (`medium.com/@aleixlopez/cloud-gpu-instances-to-solve-out-of-memory-error-2022-d5012883a272?`)

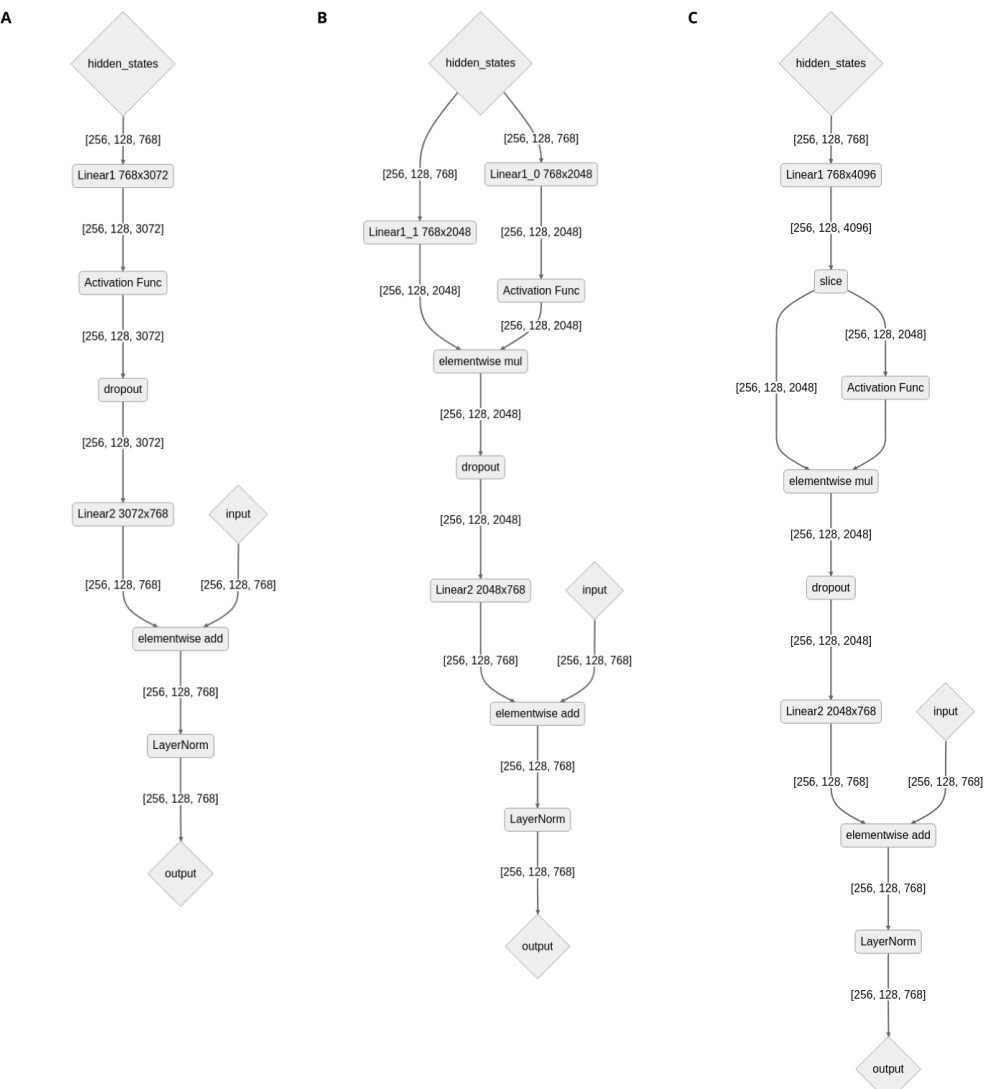

Figure S6: Standard FeedForward Transformer Block and Gated Linear Unit Modifications. Each edge shows the tensor dimension with a batch size of 256, sequence length of 128 and a hidden dimension of 768. (A) A standard transformer feedforward block. (B) Naive implementation of a Gated Linear Unit. The number of parameters in this are the same as in (A). (C) Fused implementation of a Gated Linear Unit where the two matrix multiplications (`Linear1_0` and `Linear1_1`) from (B) are fused into one (`Linear1`) with $2\times$ the parameters. Here the output is sliced, and is functionally equivalent to (B).

## J    Gated Linear Units (GLU) Optimizations

GLU adds elementwise multiplication of two linear projections and it leads to qualitative improvements over the standard Transformer block. There are multiple ways to implement GLUs and we experimented with two implementations. Figure S6 shows standard feedforward transformer block (A) and two implementations of GLUs (B-C). "Fused GLU" in (C) fuses the two matrix multiplications into one and is expected to perform better in some domains.

Figure S7 shows the performance impact of the two GLU over standard feedforward transformer block (which would be $0\%$ slowdown) for a single GPU. This figure only shows the performance of the forward pass, and the backward pass is expected to behave similarly. We can draw two conclusions

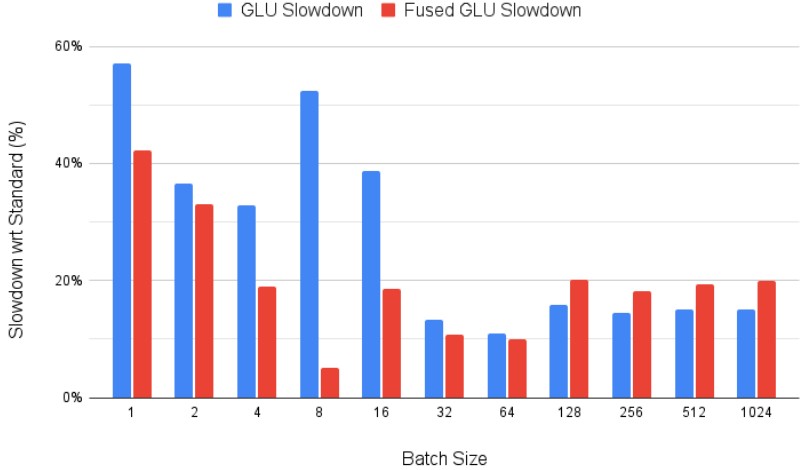

Figure S7: Slowdown of different implementations of Gated Linear Unit. This slowdown is with respect to standard feedforward transformer block. The number of parameters between standard feedforward transformer block and the two GLU implementations are the same.

from this chart: 1) For smaller batch sizes, both GLU implementations add significant overhead over the standard block 2) For batch sizes $< 128$, Fused GLU implementation is better than regular GLU implementation and beyond 128 it's slightly worse. The implementation used in the main text is the "Fused GLU" implementation (C) with global batch size 4096. Since the profiling in Figure S7 is per GPU, this is in the regime of $4096/8 = 512$.

The main reason for slowness of GLUs over standard block is extra elementwise multiplication in GLU layers. As for why fused implementation is slower, profiling analysis shows that the Linear layer ends up calling different CUDA kernels for matrix-multiplications and their relative performance varies for different sizes. While the MosaicBERT architecture in this work uses the fused GLU implementation, the analysis here indicates that it would be slightly more efficient to use the standard GLU implementation instead.

# K    Limitations and Broader Impact

## K.1    Limitations

While we trained two different model sizes, we have not pretrained a MosaicBERT model in the $> 1B$ parameter range. In this regime, it is possible there will be training stability issues; this is an area of future work.

We also only trained models for 70,000 steps and 178,000 steps of batch size 4096. It is possible that some of the Pareto properties change in the longer regime, although we suspect that this is unlikely.

## K.2    Broader Impact

BERT models are highly used for NLP tasks. By open-sourcing this work, we hope that our code and models will be used by the wider research community. We recognize however that models like BERT and MosaicBERT are tools that can be used for nefarious purposes, and that biases inherent in the training data can be reflected in the final model artefacts.

