# OpenReview forum: "MosaicBERT: A Bidirectional Encoder Optimized for Fast Pretraining"
_NeurIPS.cc/2023/Conference — NeurIPS 2023 poster_

### Official Review · Reviewer_VemN · 2023-06-08

**Soundness:** 3 good
**Presentation:** 4 excellent
**Contribution:** 4 excellent
**Rating:** 7
**Confidence:** 4

**Summary:**

This work proposes RapidBERT to train BERT in a faster way. Different from the previous accelerated method, this work attempts to employ some recent popular transformer architecture efficient designs as the basic modification of RapidBERT architecture, such revisions including the introduction of FlashAttention, AliBi, Bf16, GLU, Low Precision LayerNorm, etc. The training dataset is C4. It cost 1.13 hours on 8 A100-80G GPU.
Results on the GLUE NLU benchmark show that under fair comparison except for the training step, RapidBERT can get efficient training FLOPs compared to previous methods.

**Strengths:**

- This work employs some recently released transformer-efficient architectures in BERT training, getting RapidBERT.
- The RapidBERT got similar NLU ability compared to the vanilla-BERT.

**Weaknesses:**

- For a fair comparison, The authors should also post the RapidBERT results pre-trained on the English Wikipedia and the Books Corpus. (The original BERT study trained on English Wikipedia and the Books Corpus).
- Similar studies have been proposed in the same faster BERT pertaining track, it would be better to have a comprehensive comparison with them in detail to help readers learn more about this field.
- In line 176, this work says that "we modify the vocab size from 30522 to 30528" inspired by MEGATRON so that the vocab size is a multiple of 64, and leads to a non-trivial throughput speedup. However, in MEGATRON's work, they pad the vocab because of the 8-way model parallelism. But this work does not employ any tensor parallelism or pipeline parallelism, could you explain why such an operation can lead to a speedup in the training of RapidBERT?
- Most of the new techs (eg. FlashAttention) tried in this work have already been employed and integrated into the BERT pre-trained stage in some frameworks, e.g., Megatron, DeepSpeed [1][2]. In this manner, this work is not so innovative and may not be the first work to do this, it would be better to discuss differences with these similar works. (Actually, it is inspiring to see more works on efficient LM pertaining field, so this term is not very influential, I just brought it up.)

[1] https://github.com/NVIDIA/Megatron-LM

[2] https://github.com/microsoft/DeepSpeed

**Questions:**

Except for the adding modules (eg. FlashAttention, AliBi, etc.), there are also some recently released efficient transformer architectures (eg, RoPE, DeepNorm, etc), have the authors try them in RapidBERT. Could such methods help accelerate the training period further? It would be interesting to see if these methods could help.

**Limitations:**

Similar studies have been proposed in the same faster BERT pertaining track, this work does not have too much comparison with them in detail.

---

> ### Author Rebuttal · Authors · 2023-08-10
>
> We thank the reviewer for their detailed comments.
>
> > The authors should also post the RapidBERT results pre-trained on the English Wikipedia and the Books Corpus
>
> The primary focus of our work was to show that certain architectural modifications and training choices lead to both a speed up as well as an improvement in downstream accuracy. We were less concerned with the well-worn path of “beating” the GLUE accuracy of the original BERT, or with getting the highest GLUE scores (which requires larger models trained for much longer). Even if we had chosen to train on 40 epochs of English Wikipedia and Books Corpus, it would be difficult to benchmark directly against the original BERT, which had many differences. For example, our BERT baseline and many other BERT variants such as RoBERTa include a MLM pretraining objective instead of MLM + NSP, a larger batch size (the original paper used a batch size of 256), etc.
>
> We chose to run all experiments on the same, high quality, contemporary dataset (C4) so that we could explore the combined effects of architectures and training configuration (as opposed to data quality).
>
> > Similar studies have been proposed…comprehensive comparison with them in detail to help readers learn more about this field.
>
> We have included an extended comparison with similar studies in the updated manuscript.
>
> > In line 176, this work says that "we modify the vocab size from 30522 to 30528" inspired by MEGATRON so that the vocab size is a multiple of 64, and leads to a non-trivial throughput speedup...could you explain why such an operation can lead to a speedup in the training of RapidBERT?
>
> This was simply an empirical observation that we made when benchmarking our models, even in the absence of model parallelism. GPUs compute matrix multiplication by breaking matrices into tiles, and padding the vocab size aligns the matrix dimensions to a multiple of a fast tile size. CUDA has a heuristic which it uses to pick what tile size it uses for matrix multiplication, and it often chooses a suboptimal one when there is a “weird” matrix side length. Because of this, we and others have found that manually padding matrix dimensions can lead to non-trivial performance gains.
>
> > it would be better to discuss differences with these similar works
>
> We thank the reviewer for this comment, and have updated our manuscript to include a discussion of DeepSpeed and MEGATRON
>
> > some recently released efficient transformer architectures (eg, RoPE, DeepNorm, etc)
>
> We have a growing list of proposed efficiency improvements to transformer architectures, and are excited to explore this in future work.

---

### Official Review · Reviewer_WcGd · 2023-07-05

**Soundness:** 2 fair
**Presentation:** 2 fair
**Contribution:** 2 fair
**Rating:** 3
**Confidence:** 3

**Summary:**

The paper benchmarks several architectural changes to BERT that allows for more efficient pretraining. More specifically, the paper adds flash attention, ALiBi position representations, and GLU activations to the original BERT architectures. For fair comparison, they re-implement the baseline BERT-base using the same pretraining hardwares. They show that with these architecture modifications they can achieve better performance on downstream tasks while using smaller amounts of pretraining time. Their base model’s performance is similar to BERT-base in (Devlin et al., 2018) using approximately 1 hour of pretraining on 8 A100 GPUs, costing $20 on a standard cloud provider. They also benchmark the training cost of each architecture modification, showing that reducing the sizes of GLU matrices help to improve throughput of training significantly

**Strengths:**

The paper provides empirical evidence that combining flash attention, ALiBi position representations, and GLU activations reduces 50% of the training cost. Their ablation on GLU will benefit researchers working on the relevant areas

**Weaknesses:**

The contribution of this paper is limited. The proposed architecture modifications are all adopted from prior work. Their primary findings mostly come from tuning the hyperparameters of GLU. And their current findings depend on multiple architecture modifications, which is complicated and makes their scientific findings unclear. I’d hope that the authors can provide more experiments showing why all these modifications are necessary and whether it’s possible to simplify it further for a clearer takeaway message


**Questions:**

1. What is “Pareto Optimal”? This term is mentioned multiple times but never formally defined in the paper
2. There appears to be a throughput difference in Figure 5 between “lpLN+GLU+ALBi” and “GLU 3072”. What leads to the difference?

**Limitations:**

The authors have discussed the limitations of their work.

---

> ### Author Rebuttal · Authors · 2023-08-10
>
> > The contribution of this paper is limited.
> > I’d hope that the authors can provide more experiments showing why all these modifications are necessary.
>
> We have added many additional ablations to demonstrate the individual contributions of each of the changes to the architecture in the Author Rebuttal PDF. This is our best attempt to address the reviewer’s feedback. However, this feedback is vague and unactionable; we hope the reviewer will give us concrete and constructive feedback after the author response period so that we can improve our paper.
>
> > The contribution of this paper is limited. The proposed architecture modifications are all adopted from prior work.
>
> We strongly but respectfully disagree with this comment. The literature is full of papers that propose a single, novel improvement to an architecture like BERT (e.g., the many papers we cite and build on). However, there is very little work assessing what this collection of papers actually amounts to: whether these improvements fit together and combine to produce something even stronger, or whether they are different, mutually exclusive paths to the same gains. We believe that it is crucial to take stock of scientific progress in the manner that we have in our paper. We hope the reviewer recognizes the importance of this kind of contribution.
> > their current findings depend on multiple architecture modifications, which is complicated and makes their scientific findings unclear.
> > simplify it further for a clearer takeaway message
>
> We strongly but respectfully disagree with the reviewer’s assumption that there should be a simple, straightforward result. None of the neural network architectures we train today are simple; each comprises a multitude of small design choices and tweaks that combine to lead to a strong result. ResNets, for example, rely on a combination of residual connections, batchnorm, global average pooling, specific initialization schemes, and particular sets of data augmentation. The goal of our paper is establishing a new baseline for future work on improving BERT, and that baseline must combine all of the best techniques available. Our proposed baseline is no more complicated than any other state-of-the-art architecture in use today.
>
> > Their primary findings mostly come from tuning the hyperparameters of GLU.
>
>
>
> Our primary contribution is building and extensively benchmarking one of the fastest architectures for BERT pretraining. We believe this is an important contribution to the academic and engineering communities, as fast, cheap pretraining enables ML practitioners.
>
> We believe that the reviewer has misunderstood our primary contribution to be the change of the dimension of the intermediate layer for the Gated Linear Unit.
>
> > What is “Pareto Optimal”?
>
> We thank the reviewer for bringing to our attention that we do not explicitly define the term “Pareto optimal,” and have fixed this in the updated manuscript.
>
> In order to make meaningful improvements in neural network training efficiency, ML practitioners
> must be able to compare between different choices of network architectures, hyperparameters, and training algorithms. One straightforward way to do this is to characterize the tradeoff between accuracy and training time with a “tradeoff curve” or a “Pareto curve.” Pareto curves can be generated by varying the length of training for each model configuration; longer training runs take more time but tend to reach higher quality. For a fixed model and task configuration, this method of generating tradeoff curves is an estimate of the theoretical Pareto frontier, i.e. the set of all of the best possible tradeoffs between training time and accuracy, where any further attempt to improve one of these metrics worsens the other.
>
> We therefore consider one model Pareto optimal relative to another if it has superior accuracy across different training budgets while keeping everything else fixed. Many studies advertise novel architecture approaches without measuring wall clock time, or show an improvement for a single training duration. In our paper we show that RapidBERT-Base is Pareto optimal relative to our BERT-Base baseline for both short training durations and long training durations (Figure 2). We also show that BERT-Large and RapidBERT-Large are not Pareto optimal relative to BERT-Base and RapidBERT-Base for training durations under ~30 hours.
>
> > throughput difference in Figure 5
>
> In Figure 5A, “lpLN+GLU+ALiBi” is bert-base with the additional pieces of low precision LayerNorm, GLU and ALiBi on top of the classic BERT architecture. In Figure 5B, “GLU 3072” indicates the complete RapidBERT architecture, which includes full-sized GLU, unpadding, FlashAttention, optimal vocab size, masked language modeling ratio of 30%, etc. We indicated this in the caption for Figure 5B “Throughput of the “complete” RapidBERT-Base with different intermediate sizes for GLU,” but will elaborate on this to be more clear.

---

> ### Comment · Area_Chair_pV83 · 2023-08-16
> **Check rebuttal**
>
> @Reviewer WcGd,
>
> Does the rebuttal address your concerns? Could you read it and update your review accordingly?

---

### Official Review · Reviewer_NjCS · 2023-07-07

**Soundness:** 4 excellent
**Presentation:** 4 excellent
**Contribution:** 2 fair
**Rating:** 7
**Confidence:** 3

**Summary:**

The paper introduces RapidBERT, an architecture and training paradigm for pretraining BERT-style language models that is cost-effective. The proposed approach incorporates several modifications into the conventional transformer encoder block, including FlashAttention, Attention with Linear Biases, Gated Linear Units, Unpadding Module, and a low precision LayerNorm. The pretraining process avoids the Next Sentence Prediction task and follows the RoBERTa practices, using the C4 dataset with a 30% masking ratio in MLM.

RapidBERT demonstrates faster convergence and achieves a better accuracy versus time Pareto curve during pretraining. Their base model consistently outperforms the original BERT model on average across the GLUE dev tasks. Remarkably, the training process only takes 1.13 hours using 8 A100 GPUs, resulting in a cost of approximately $20.

The authors show that RapidBERT is Pareto-optimal when compared to BERT for both base and large models. They also highlight the necessity of extensive training for larger models, as RapidBERT-Base outperforms both BERT-Large and RapidBERT-Large for a significant portion of their training.

In addition, the paper includes a comprehensive ablation analysis of the design choices made in RapidBERT and evaluates the throughput of each architecture. The results indicate that the GLU modification leads to a decrease in throughput, while ALiBi has minimal impact on throughput. On the other hand, incorporating low precision LayerNorm significantly improves throughput.

Overall, the paper presents RapidBERT as an efficient and effective approach for pretraining BERT-style language models, showcasing its superior performance, cost-effectiveness, and the benefits of the proposed modifications.

**Strengths:**

- The paper introduces a time and compute-efficient approach for pretraining BERT-like models, offering significant advantages over previous works. This approach holds promise for pretraining task-specific language models that do not require high parameter requirements, making it highly practical and valuable in various applications.
- The implementation details provided in the paper are clear and comprehensive, contributing to the reproducibility of the research. The authors effectively explain the concepts and methodologies, ensuring a thorough understanding of the proposed approach.
- The research conducted on related work demonstrates a comprehensive exploration of the existing literature. The design choices made in this study are grounded in well-established prior works, and the ablation study further strengthens the validity and effectiveness of the proposed modifications.
- The paper presents an intriguing finding regarding the impact of model size on performance in specific domains. The authors discover that larger models may not always yield superior results due to limited data availability and increased compute time. This insight, demonstrated by the Pareto optimality of RapidBERT-Base over BERT-Large and RapidBERT-Large for a significant portion of the training, adds a valuable contribution to the understanding of model performance and scalability.

**Weaknesses:**

~- The approach primarily combines existing techniques, such as FlashAttention, GLUs, and low precision LayerNorm. While the authors introduce the novel aspect of maintaining a 30% masking ratio, the overall novelty of the architectural choices is limited.~
~- It would have been beneficial to include a discussion comparing the proposed approach with the original RoBERTa model as a baseline. This comparison would provide a clearer understanding of the improvements achieved by RapidBERT and highlight its unique contributions.~
- The paper lacks a discussion on how the architectural choices made in RapidBERT could facilitate the development of larger language models, such as GPT 3.5. Exploring the potential scalability and benefits of these choices for larger models would enhance the practical relevance and implications of the research.
- The ablation studies do not investigate the effects of changing the masking ratio. Including an analysis of different masking ratios would offer insights into the choice behind keeping the ratio to be 30%.
- The paper would benefit from discussions on the comparison with RoBERTa, the scalability of the approach to larger models, and the effects of different masking ratios in the ablation studies.

**Questions:**

- Can you provide a rationale for considering BERT as a "strong" baseline in your study? Why was RoBERTa not chosen as the baseline for comparison, and what factors led to the selection of BERT as the benchmark model?
~- It would be helpful to gain insights into the absence of numerical instabilities in bfloat16. Could you provide any intuition or theoretical reasoning behind why it doesn't happen in your case?~

**Limitations:**

The discussion on limitations is satisfactory. There are some limitations discussed in the appendix regarding potential training stability issues on the larger models. Potential negative societal impacts (ease of pre-training could lead to ease of more biased/inappropriate models) is also discussed in broader impacts section. General limitations with any pre-trained LMs will apply to RapidBERT.

---

> ### Author Rebuttal · Authors · 2023-08-10
>
> We thank the reviewer for their detailed comments and suggestions.
>
> > approach primarily combines existing techniques…the overall novelty of the architectural choices is limited
>
> We strongly but respectfully disagree with the reviewer’s concerns about novelty. The literature is full of papers that propose a single, novel improvement to an architecture like BERT. (E.g., the many papers we cite and build on.) However, there is very little work assessing what this collection of papers actually amounts to: whether these improvements fit together and combine to produce something even stronger, or whether they are different, mutually exclusive paths to the same gains. We believe that it is crucial to take stock of scientific progress in the manner that we have in our paper, and we emphasize the novelty of (1) the nontrivial work we needed to perform to get these many improvements to work well together and (2) the fact that we sought to combine many existing methods rather than invent yet another new technique of uncertain value. We hope the reviewer recognizes the importance of this kind of contribution and the novelty inherent in performing this work.
>
> > It would have been beneficial to include a discussion comparing the proposed approach with the original RoBERTa model as a baseline
>
> We appreciate the reviewer’s comments regarding RoBERTa, and have elaborated on this in the updated version of the manuscript. RoBERTa is a “training recipe” for BERT that preserves the network architecture but changes hyperparameters and training data. The RoBERTa paper showed that the training dataset is incredibly important; for example, RoBERTa trained on 350GB of data while BERT only trained on 16GB of data. The RoBERTa paper also showed that the next sentence prediction (NSP) objective was superfluous, and showed that increasing the batch size led to accuracy gains. These RoBERTa training choices have become the standard in the field, and our general training recipe choices resemble RoBERTa more than the original BERT (as we mention in the paper).
>
> Since our goal was to show that both our architectural and training recipe choices led to strict accuracy vs. wall clock time Pareto improvements, we wanted to make sure that we were comparing our models with a very strong baseline. Therefore both BERT and RapidBERT used the same dataset (C4), tokenizer, learning rate schedule, batch size and device microbatch size, hardware, etc. The initial untrained BERT architecture and the RoBERTa architecture are the same, so the setup would be the same for our strong baseline. We have included more reported RoBERTa values in the Appendix of our updated manuscript as a helpful reference point.
>
> > how the architectural choices … could facilitate the development of larger language models
>
> We thank the reviewer for making this point, and have expanded on this in our updated manuscript. All of the architectural and configuration choices made in RapidBERT can in principle be applied to larger models, although they might have different effects on throughput. Additionally, they should not pose any issues with different forms of parallelism (some of the modifications we explored were used successfully at large scale in frameworks like MEGATRON). For example, Gated Linear Units should have no issues with tensor parallelism as long as the matrix multiplications are a multiple of the TP world size. As models are scaled, LayerNorm becomes a smaller portion of the total compute, so low precision LayerNorm might not matter as much.
>
> > The ablation studies do not investigate the effects of changing the masking ratio
>
> Multiple studies have shown that the change to the MLM masking ratio from 15% (in the original BERT) to 30% leads to a small but significant improvement in downstream GLUE performance. In “Should You Mask 15% in Masked Language Modeling?” Wettig et al. find that constant MLM masking ratios above 15% lead to improved average GLUE and SQuAD scores for bert-base and bert-large. Similarly, in the recently published paper “Dynamic Masking Rate Schedules for MLM Pretraining,” Ankner et al. find that a constant MLM masking ratio of 30% consistently outperforms a MLM masking ratio of 15% for BERT-base. We have elaborated on this in the updated manuscript.
>
> We have also included an updated figure in the Author Rebuttal PDF showing that an MLM ratio of 30% leads to a small improvement in the downstream GLUE score over MLM 15% without affecting the wall clock time.
>
> > the absence of numerical instabilities in bfloat16.
>
> As we were prototyping our RapidBERT architecture and training recipe, we were surprised to find that low precision (bfloat16) LayerNorm afforded significant speedups without compromising stability. While instabilities such as loss spikes are strongly coupled to the learning rate schedule and other hyperparameters, we found low precision LayerNorm to be a strict Pareto improvement when benchmarking wall clock time and downstream accuracy in our hyperparameter regime. We don’t have theoretical intuitions for why numerical instabilities didn’t occur in our hyperparameter regime. We should note however that our learning rate schedule was not overly aggressive (warmup followed by linear decay with peak values well within the range of published studies), and that this could have mitigated loss spikes. Finally, we should also note that the NVIDIA Apex library and Megatron both use a form of low precision LayerNorm in their code, but it is not documented in any papers that we could find. We have elaborated on this in our updated manuscript.

---

### Official Review · Reviewer_xi9s · 2023-07-07

**Soundness:** 3 good
**Presentation:** 2 fair
**Contribution:** 2 fair
**Rating:** 6
**Confidence:** 5

**Summary:**

This paper proposes a new efficient recipe for training BERT, matching the original performance of BERT on GLUE in ~1h on 8x A100. To do so, the authors leverage a number of architectural/implementation improvements: FlashAttention, ALiBi, GeGLU, `bf16` layer norm, unpadding, and tweaks to the masking ratio. The authors find that their recipe is optimal even when considering longer runs, and that it transfers well to BERT-Large.

**Strengths:**

* **S1.** Reducing the costs associated with training language models can help practitioners iterate faster, improving downstream research outcomes. This makes this paper potentially valuable to the community, as it improves upon previously introduced similar recipe such as CrammingBERT.

* **S2.** The authors detail their contributions and open-source their code, making these results reproducible and enabling the community to build upon them.

* **S3.** The authors feature an updated baseline (BERT-Base) that helps for fair comparisons in their setup.

**Weaknesses:**

* **W1. It is difficult to untangle individual contributions to the final result.**
    * **W1.1.** The proposed recipe visibly has some positive impact on GLUE score, as Figure 3 shows that it achieves significantly better performance than BERT-Base after the same amount of training. However, that impact is never quantified in the paper. Here, the ablations should not only focus on throughput, but also on how the proposed interventions might impact the GLUE score.
   * **W1.2.** The proposed baseline is very strong (which is a positive point), and it would be good in Table 1 to also showcase the time it take for it to reach a 79.6 GLUE score. Furthermore it would be interesting to identify what makes the baseline such a strong one (with a final score much higher than BERT-Base). Is it the change in data to C4 (which authors acknowledge as an important factor l161)? Something else?
   * **W1.3.** l112 the authors discuss using ALiBi to pretrain with a shorter sequence length and extrapolate at test time. It's unclear if this end up being used, and if it is included in the ALiBi ablations.
   * **W1.4.** l251 it is disappointing to not ablate adequately every component, especially since the value of this paper lies in having a potentially systematic approach to performance improvements of BERT models.

* **W2. The Pareto frontiers described may be slightly misleading, as they do not account for LR schedule.** The so-called Pareto optimality is obtained by taking points from the same run, instead of having one run per pretraining budget on the plot. This approximation penalises intermediate budgets, as they are evaluated with an incomplete LR schedule. While I don't think this has a significant influence in this work, since the authors discuss Pareto optimality so much, this should at the very least be clearly discussed as a limitation to avoid misleading other authors. This issue is particularly relevant, as it has lead to significant misunderstandings around scaling laws for instance (see Hoffmann et al., 2022).

* **W3.** The paper feels a bit repetitive, as if it had been stretched to fit the 9 pages of content. Section 3.2 and 3.3 are particularly egregious in this regard, and more time could instead be spent on ablations.

* **W4.** (minor nits) l98 reference to Triton should cite "Triton: An Intermediate Language and Compiler for Tiled Neural Network Computations" (Tillet et al., 2019); l145 the sentence "results from NVIDIA and others" is confusing, as the final work cited is not from NVIDIA -- there should be a citation somewhere for the NVIDIA results.

**Questions:**

This has the potential to be a valuable paper to the community, by systematically identifying and detailing interventions that can accelerate the training of large language models like BERT. Reducing iteration costs is in particular a great enabler for researchers. Unfortunately, the lack of systematic ablations on GLUE score and the limited throughput ablations make this paper fall somewhat short of its promise. Accordingly, I am rating it as a **Borderline Reject (4)** but would be willing to increase my score to an accept should some of my concerns be addressed.

**EDIT: following rebuttal, I have updated my score to a Weak Accept (6).**

* **Q1.** Could the authors explain why Figure 3 shows a significant improvement in GLUE score for RapidBERT over BERT? Would it be possible to ablate for this improvement, to ultimately better understand which intervention improves throughput and which improves "modelling performance" (i.e., GLUE score)?

* **Q2.** Could the authors explain why the baseline used is so much above the original BERT-Base?

* **Q3.** Could the authors clarify whether they use for RapidBERT a shorter training seqlen to accelerate training as proposed?

* **Q4.** Could the authors provide further performance ablations for FlashAttention/unpadding?

* **Q5.** (more of a suggestion) Could the authors better describe the limitation of their approach regarding Pareto-optimality?

Small suggestions:
* l13/14 in the abstract "When pretrained from scratch on the C4 dataset, this base model achieves the downstream average GLUE score of 79.6 in 1.13 hours on 8 A100 80 GB GPUs at a cost of roughly USD 20" it would be good to mention this is the GLUE score achieved by the original BERT; for instance "RapidBERT achieves the same downstream average GLUE score as the original BERT (79.6) in 1.13 hours on 8 A100 80 GB GPUs at a cost of roughly USD 20".

**Limitations:**

While there is no explicit section, some limitations in terms of scope are discussed in the conclusion.

---

> ### Author Rebuttal · Authors · 2023-08-10
>
> We thank the reviewer for their detailed comments and suggestions, and we try to address them here.
>
> **W1** One of the primary goals in this paper is to develop a high-performance BERT architecture and recipe for pretraining on commercially available high-end hardware (A100-80GB GPUs). We agree with the reviewer that ablations should focus on downstream performance in addition to throughput, and have included updated plots with the individual effects of GLU, low precision LayerNorm, etc. on both wall clock time and GLUE scores.
>
> **W1.1 (How ablations affect GLUE score)** We ran further ablations in response to the reviewer’s comments and have plotted downstream GLUE accuracy as a function of measured pretraining wall clock time. We provide figures in the Author Rebuttal PDF. The patterns here shed light on the individual effects of various architectures (e.g. BERT+GLU, BERT+low precision LayerNorm) and training configurations (e.g. BERT + 30% masking ratio). On average, all methods seem to provide a slight accuracy boost to BERT-base. Increasing the masking ratio to 30% leads to a slight accuracy boost while not affecting the WCT, while turning off dropout in the attention layer leads to a slight improvement in both accuracy and WCT. Low precision LayerNorm leads to a significant speedup. Gated Linear Units (GLU) add more parameters to the model and lead to a significant slowdown while providing an accuracy boost. As a point of reference, we also benchmark the full RapidBERT as well as RapidBERT without FlashAttention and with dropout set to 0.1 (the standard BERT-base  configuration).
>
> **W1.2 (Strong baseline)** Thank you for this suggestion. We will include the time it takes for our strong baseline to reach a 79.6 GLUE score in Table 1. We are confident that the C4 dataset is the main reason for the strong BERT baseline relative to the original BERT (Devlin et al. 2018). The original BERT was trained for 40 epochs on Wikipedia + Books Corpus (16GB of text for a single epoch). C4 contains Wikipedia and Books Corpus, as well as much more data (roughly 350 GB). Besides the use of C4, we note some other differences from the original BERT paper: 8xA100 80 GB GPUs, MLM pretraining objective instead of MLM + NSP, a sequence length of 128 throughout the entirety of pretraining (instead of starting with 128 and then expanding to 512 towards the end of pretraining), larger batch size (the original paper used a batch size of 256), an increased vocab size of 30,528 (the original used a vocab size of 30,000, while the default Hugging Face vocab size for the bert-base-uncased tokenizer is 30,522), and the use of PyTorch 1.13. We used the original bert-base-uncased tokenizer for all our models; it is likely that a more modern tokenizer would improve GLUE results as well.
>
> **W1.3 (ALiBi)** Our main motivation for including ALiBi was to allow for long sequence extrapolation and-or finetuning in downstream use cases. There is increasing demand in the ML community for long context windows and alternatives to positional embedding schemes. We agree with the reviewer that our submission did not decisively show that ALiBi extrapolates well at test time. Both BERT and RapidBERT pretraining was done on a sequence length of 128, while finetuning was done for a sequence length of 512 (and have made this clear in the updated manuscript). Unfortunately the GLUE benchmark does not have many examples with long sequence lengths above 512 tokens, and is therefore not a great benchmark for testing sequence length extrapolation.
>
> **W1.4 (Ablations)** We agree with the reviewer that ablations here should not only focus on throughput but also downstream GLUE accuracy. We have provided updated experiments on GLUE score vs. wall clock time for BERT, BERT+GLU, BERT+low precision LayerNorm, BERT+ mlm 30% ratio, BERT+no attention dropout, RapidBERT, and RapidBERT-FlashAttention in the Author Rebuttal PDF. In the final version of the manuscript, we will also include results for BERT+ALiBi, BERT+ALiBi+GLU, etc.
>
> **W2 (Pareto Curve)** The Pareto plots in Figures 1-4 are constructed by taking points from the same runs (with the same learning schedule), instead of having one run per pretraining budget on the plot. We did this for cost reasons; it would have been too expensive to do every run separately. We emphasize that this approximation penalizes intermediate budgets, as they are evaluated with an incomplete LR schedule, so we expect that our results would be even stronger had we used the learning rate schedule corresponding to the training time for each point.
>
> **W3 (Repetitive)** We appreciate this feedback, and we will cut some of the unnecessary language from sections 3.2 and 3.3
>
> **W4 (References)** Thank you for these suggestions. We have updated the manuscript to properly cite the Triton paper and the correct NVIDIA reference for unpadding (which was NVIDIA’s MLPerf v1.1 submission).
>
>
>
>
>
> **Q1:** We hope that the additional ablation experiments shed light on the large gap between RapidBERT and BERT in Figure 3. Each of the individual architecture modifications lead to either a downstream accuracy boost or a speedup; combined (RapidBERT) they lead to the full Pareto improvement over BERT.
>
> **Q2:** We have addressed this question in comment W1.2
>
> **Q3:** We pretrain with sequence length 128 for both the baseline BERT and for RapidBERT. We therefore do not count this as a speedup method. We then finetune both BERT and RapidBERT with sequence length 512.
>
> **Q4:** We have provided some FlashAttention ablations in the Author Rebuttal PDF.
>
> **Q5:** We address the question of Pareto optimality in W2

---

> > ### Comment · Reviewer_xi9s · 2023-08-16
> > **Answer to the rebuttal**
> >
> > First, I would like to thank the authors for providing an extensive rebuttal to each reviewer, as well as additional results.
> >
> > I believe that this additional information+discussions significantly improve the quality and value of the paper. I will trust that the authors update their paper based on the rebuttals they have provided.
> >
> > Accordingly, I have updated my score to a **Weak Accept (6)**.
> >
> > Re: as a small clarification on W2, I agree that this does not impact the results presented -- I just pointed to the need for mentioning this explicitly (that intermediary results are penalised), as I believe this is an important point of confusion in the community.

---

> > > ### Author Response · Authors · 2023-08-16
> > > **We thank the reviewer for their comments and appreciate the updated score.**
> > >
> > > We thank the reviewer for their comments and appreciate the updated score.
> > >
> > > We agree with the reviewer that the nuances of learning rate schedules and Pareto curves are an important point of confusion in the community, and will make sure to discuss this explicitly in the final version of the paper.

---

### Official Review · Reviewer_Ss7L · 2023-07-20

**Soundness:** 3 good
**Presentation:** 3 good
**Contribution:** 3 good
**Rating:** 7
**Confidence:** 4

**Summary:**

This paper presents a training recipe that can train a BERT-style encoder model efficiently (1.13 hours on 8 GPUs). The recipe combines several techniques including a higher masking rate for MLM, bf16, optimized vocabulary size. The model is trained on the C4 dataset for 1.13 hours and achieves 79.6 on GLUE. The paper conducts ablation studies to characterize properties of the architecture design choices.

**Strengths:**

* The paper presents very impressive results of pre-training a BERT-like model very efficiently without loss in accuracy. In general, I believe that this is a solid outcome and it is definitely a good addition to the community, as it can enable more researchers to pretrain custom BERT models from scratch.
* The paper presents ablation experiments that quantify the impact of each proposed change.
* The paper is well written and easy to follow. Replicating the results should not be hard.


**Weaknesses:**

* The experiments are only conducted on GLUE tasks; it is not clear how well the trained model transfers to other datasets.
* There is no fair comparison (i.e., under the same hardware setup) of the proposed recipe and other efficient training methods such as CrammingBERT.


**Questions:**

* The paper uses a 30% MLM masking ratio for RapidBERT instead of 15% for BERT. How do you compare the improvement from this change to the gains from your architectural modifications?

**Limitations:**

I didn’t see a clearly potential negative societal impact of this paper.

---

> ### Author Rebuttal · Authors · 2023-08-10
>
> We thank the reviewer for their comments and questions.
>
> > The experiments are only conducted on GLUE tasks; it is not clear how well the trained model transfers to other datasets.
>
> We chose to focus exclusively on the GLUE benchmark for multiple reasons. Since GLUE is a classic benchmark, fine-tuning on GLUE makes it easy to compare results across both recent and more “classic” papers (e.g. the original BERT). On a more practical level, the data presented in our paper represents hundreds of GPU hours, and we had to make the pragmatic decision to extensively benchmark on one dataset instead of benchmarking on multiple datasets in a limited way.
>
> > There is no fair comparison (i.e., under the same hardware setup) of the proposed recipe and other efficient training methods such as CrammingBERT.
>
> While some of the architecture modifications in our paper are shared with Cramming BERT, including FlashAttention and Gated Linear Units (GLU), we consider our work complimentary in the following ways:
> * We try to build and benchmark the fastest pretraining recipe with the highest performance on top-end hardware, while Cramming BERT focuses on the low-resource scenario (one consumer GPU for one day). In that sense, it focuses on one lower-performance point on the pareto frontier, whereas we focus on the entire pareto frontier.
> * We evaluate multiple model sizes (i.e. both RapidBERT-Base and RapidBERT-Large), while Cramming BERT only analyzes BERT-Base
> * We share full Pareto curves for BERT Base and Large as well as throughput and accuracy ablations (see Author Rebuttal PDF for further ablations). We believe this extensive data will be exceedingly useful to practitioners. We have already open-sourced our code and hope that the results from hundreds of GPU hours will be useful to the ML community
>
> > The paper uses a 30% MLM masking ratio for RapidBERT instead of 15% for BERT. How do you compare the improvement from this change to the gains from your architectural modifications?
>
> Multiple studies have shown that the simple change to the MLM masking ratio from 15% (in the original BERT) to 30% leads to a small but significant improvement in downstream GLUE performance. In “Should You Mask 15% in Masked Language Modeling?” Wettig et al. find that constant MLM masking ratios above 15% lead to improved average GLUE and SQuAD scores for bert-base and bert-large. Similarly, in the recently published paper “Dynamic Masking Rate Schedules for MLM Pretraining,” Ankner et al. find that a constant MLM masking ratio of 30% consistently outperforms a MLM masking ratio of 15% for bert-base. We have elaborated on this in the updated manuscript.
>
> We have also included an updated figure in the Author Rebuttal PDF showing that an MLM ratio of 30% leads to a small improvement in the downstream GLUE score over MLM 15% without affecting the wall clock time.

---

### Author Rebuttal · Authors · 2023-08-10

We ran further ablations in response to reviewer comments and have plotted downstream GLUE accuracy as a function of measured pretraining wall clock time. The comments here describe the additional experiments in detail.

The patterns in Figures R1 and R2 shed light on the individual effects of various architectures (e.g. BERT+GLU, BERT+low precision LayerNorm) and training configurations (e.g. BERT + 30% masking ratio). On average, all methods seem to provide a slight accuracy boost to BERT-base. Increasing the masking ratio to 30% leads to a slight accuracy boost while not affecting the WCT, while turning off dropout in the attention layer (BERT+drpt=0) leads to a slight improvement in both accuracy and WCT. Low precision LayerNorm (BERT+lpLN) leads to a significant speedup (i.e. a shift to the left). Gated Linear Units (BERT+GLU) add more parameters to the model and lead to a significant slowdown while providing an accuracy boost. As a point of reference, we also benchmark the full RapidBERT as well as RapidBERT without FlashAttention and with attention dropout set to 0.1 (the standard BERT-base  configuration).

All BERT-Base models here were pretrained on C4 for 70,000 steps with batch size 4096, and microbatch size 256 on 8xA100 80GB GPUs. All models were initialized with the same seed and shared all other hyperparameters including the bert-base uncased tokenizer, the learning rate schedule, AdamW as the optimizer, etc. Final pretraining checkpoints were then finetuned on GLUE following the details in the appendix of our paper. The points represented in these GLUE plots are final finetuning checkpoints.

These plots highlight the importance of benchmarking with Pareto curves, as it is not possible to tell from these plots alone whether training BERT-base for 2 more hours leads to better performance than BERT+GLU (for example).

* BERT-base: standard BERT-base (110M parameters) with attention dropout=0.1 and feedforward dropout=0.1, vocab size set to 30522, MLM=15% (all Hugging Face standard configurations)

* BERT+drpt=0: standard BERT-base, except that the attention in the dropout layer is set to 0 instead of the default 0.1

* BERT+GLU: standard BERT-base, with GLU for the feedforward component of the encoder block

* BERT+lpLN: standard BERT-base, except with low precision LayerNorm (bfloat16)

* BERT+mlm30: standard BERT-base, except with a masked language modeling masking ratio of 30%

* RapidBERT: the complete RapidBERT detailed in the paper, including GLU (where the dimension of the intermediate layer is 3072), ALiBi, low precision LayerNorm, unpadding, MLM 30%, vocab size 30528 (a multiple of 64) and the attention dropout=0.

* RapidBERT-FlashAttn+drpt=0.1: RapidBERT _without_ Flash Attention and _with_ the attention dropout=0.1

---

### Decision · Program_Chairs · 2023-09-21

**Decision:**

Accept (poster)

**Comment:**

The paper introduces RapidBERT, a BERT-style encoder architecture and training recipe optimized for fast pretraining in NLP research. Incorporating various advanced techniques and best practices, RapidBERT achieves competitive performance with significantly reduced pretraining time and cost, e.g., the base model achieves the downstream average GLUE score of the original BERT study in 1.13 hours on 8 A100 80 GB GPUs at a cost of roughly $20 on a standard cloud provider.

Overall it is a solid work with solid and replicable results, and potential value for practitioners. Added experiments (e.g., beyond GLUE), ablations, benchmarking (w.r.t. RoBERTa and CrammingBERT), and simplification were advised to strengthen the methodology, analysis, and contributions.